# Temporal genetic association and temporal genetic causality methods for dissecting complex networks

Luan Lin[1,2], Quan Chen[1,2], Jeanne P. Hirsch[3], Seungyeul Yoo[1,2], Kayee Yeung [4], Roger E. Bumgarner[4], Zhidong Tu[1,2], Eric E. Schadt[1,2,5] & Jun Zhu [1,2,5]

A large amount of panomic data has been generated in populations for understanding causal relationships in complex biological systems. Both genetic and temporal models can be used to establish causal relationships among molecular, cellular, or phenotypical traits, but with limitations. To fully utilize high-dimension temporal and genetic data, we develop a multi-variate polynomial temporal genetic association (MPTGA) approach for detecting temporal genetic loci (teQTLs) of quantitative traits monitored over time in a population and a temporal genetic causality test (TGCT) for inferring causal relationships between traits linked to the locus. We apply MPTGA and TGCT to simulated data sets and a yeast F2 population in response to rapamycin, and demonstrate increased power to detect teQTLs. We identify a teQTL hotspot locus interacting with rapamycin treatment, infer putative causal regulators of the teQTL hotspot, and experimentally validate *RRD1* as the causal regulator for this teQTL hotspot.

[1] Department of Genetics and Genomic Sciences, Icahn School of Medicine at Mount Sinai, New York, NY 10029, USA. [2] Icahn Institute for Genomics and Multiscale Biology, Icahn School of Medicine at Mount Sinai, New York, NY 10029, USA. [3] Department of Pharmacology and Systems Therapeutics, Icahn School of Medicine at Mount Sinai, New York, NY 10029, USA. [4] Department of Microbiology, University of Washington, Box 358070, Seattle, WA 98195, USA. [5] Sema4, a Mount Sinai venture, Stamford, CT 06902, USA. Correspondence and requests for materials should be addressed to J.Z. (email: jun.zhu@mssm.edu)

Among the top objectives in modeling living systems is the construction of mathematical models capable of predicting future states of a system given a set of initial starting conditions. Whether predicting the risk of a disease at any point along one's life course given genetic, environmental, and clinical data[1], or predicting the molecular response to perturbations on a given protein or proteins[2] and the consequences of that molecular response at the cellular and ultimately physiological levels, identifying the complex web of causal relationships among molecular features and between molecular and higher-order features is central to achieving an accurate understanding of complex biological systems[3–5]. Whereas descriptive models may achieve a high degree of accuracy in classifying individuals based on any number of features (e.g., distinguishing poor from good prognosis in breast cancer based upon tumor gene expression data[6]), predictive models seek to represent causal relationships between variables of interest and as a result reflect information flow through the system, thus enabling the identification of key modulators of a given biological process[7], key points of therapeutic intervention[8], or other interesting aspects of system behavior that can aid in our understanding of it[9–11].

Building highly accurate predictive models depends on establishing causal relationships among the variables of interest. Elucidating physical interactions have been the primary means by which biologists establish causal relationships. For example, a transcription factor binding to a stretch of DNA[12] and thus facilitating the transcription of a gene that in turn activates a given biological pathway[13]. Another type of causal relationships inferred through statistical causality tests has achieved widespread utility[5,14]. This type of causal relationships is considered as a weak form of causality and experimental follow-ups are generally needed to validate them. However, this weak causality enables us to orient the vast sea of correlations observed among hundreds of thousands of molecular phenotypes that can be simultaneously assayed, according to the direction of information flow.

Methods such as Bayesian network reconstruction algorithms have been devised to infer causal relationships among correlated traits[3,15,16]. However, such methods based on correlation data alone are well known to be generally unable to uniquely resolve the causal relationships among traits, given the different types of possible relationships between traits may not be statistically distinguishable from one another (e.g., see Fig. 1a). To break this statistical symmetry so that causal relationships can be more precisely resolved, a systematic source of genetic and/or environmental perturbations must be introduced. Genetics-based causal (GC) inference anchors on the genetic locus, information can only flow from the genetic locus, so that other Markov equivalent structures are not biologically possible (Fig. 1b). GC have demonstrated widespread utility in biology over the last decade[14,17–19]. Panomic quantitative trait loci relating DNA variants to panomic data and higher-order phenotypes such as disease state have been appropriately leveraged to infer causal relationships between molecular data and higher-order phenotypes[7,20]. The successes of GC inference notwithstanding, these approaches are not without their weaknesses. For example (Fig. 1c), if two traits are related via a negative feedback loop, the sign of their correlation and the direction of the causal relationship inferred from a GC approach would be determined by the average strength of the genetic perturbations on each trait in the population[9] (the causal relationship would flow in the direction of the dominant genetic perturbation).

Similarly, a broad range of data, from imaging data to panomic and clinical data, have been scored longitudinally in populations. Time-series based causal (TSC) inference[21–24] such as dynamic Bayesian networks or Granger causality has been developed to infer causal relationships from such data (Fig. 1d). However, TSC inference often cannot resolve even simple causal relationships. For example, if a trait (gray node in Fig. 1e) causes changes in two other traits (green and blue nodes in Fig. 1e), but a longer lag for the impact of the gray node on the blue node is observed compared with the green node, then the time-series signal for the green node may well predict the behavior of the signal from the blue node, leading to a false causal inference (Fig. 1e). Both genetic and temporal data are needed to solve these problems.

To date, inferring causality by jointly considering temporal and genetic dimensions in a formal modeling framework has not been systematically explored in high-dimension omics data. Integrating these two dimensions, which have a fundamental role in enabling causal inference, has the potential to enhance the power to resolve causal relationships and to provide a more accurate view of regulatory networks in biological systems. Previous method[25] proposed to model growth-related temporal traits using a multivariate normal distribution and assumed that the mean vectors followed a logistic growth curve. In the context of temporal gene expression traits, the trajectories are usually much more complex and thus require more flexible fitting options.

Here we present a multivariate polynomial temporal genetic association (MPTGA) model that formally integrates genetic and temporal information to identify genetic association and a temporal genetic causality test (TGCT) to infer causal relationships among quantitative traits. To highlight the utility of this type of integrated tests, we apply it to transcriptomic data generated in a segregating population of yeast that were profiled at six different time points in response to treatment with the drug rapamycin. From these data, we demonstrate that the MPTGA test identifies significantly more genetic associations than the sum of the relationships identified via a genetic association test independently applied at different time points. In addition, we demonstrate that this approach has increased power to detect the causal regulators of expression quantitative trait loci (eQTL) hotspots that have been previously defined in this population, including the identification of regulators that had previously evaded direct detection. Finally, we identify and experimentally validate new causal regulators for temporal eQTL (teQTL) hotspots in this yeast population that explain the gene-by-drug interactions identified in our experiment.

## Results

**Overview of temporal genetic association and causality tests.** As living systems are dynamic, constantly changing over time to adjust to different states and environmental conditions, the extent to which different genetic loci will impact a given trait may vary over time. There are multiple ways to model the behavior of a trait over time with respect to a given genetic locus. A simple approach is to perform eQTL analysis at each time point independently, then combine the results from analysis of all time points (referred as the union method) or perform meta-analysis based on Fisher's method (referred as the Fisher's method). We can also apply multivariate analysis of variance (MANOVA) to detect the difference of gene expression levels across different time points between different genotype groups. Alternatively, we can model time-series data by different autoregressive (AR) models, then assess whether the AR models are different with regard to different genotypes (referred to as the AR model). Alternatively, we can consider a quantitative trait following a polynomial function with regard of time and then employ a straightforward regression approach to model the trait with respect to a given genetic locus (referred as the regression method). If we further assume that for each genotype the trait over time follows a multivariate normal distribution similar to Ma et al.[25] and the variances across subsequent time points are

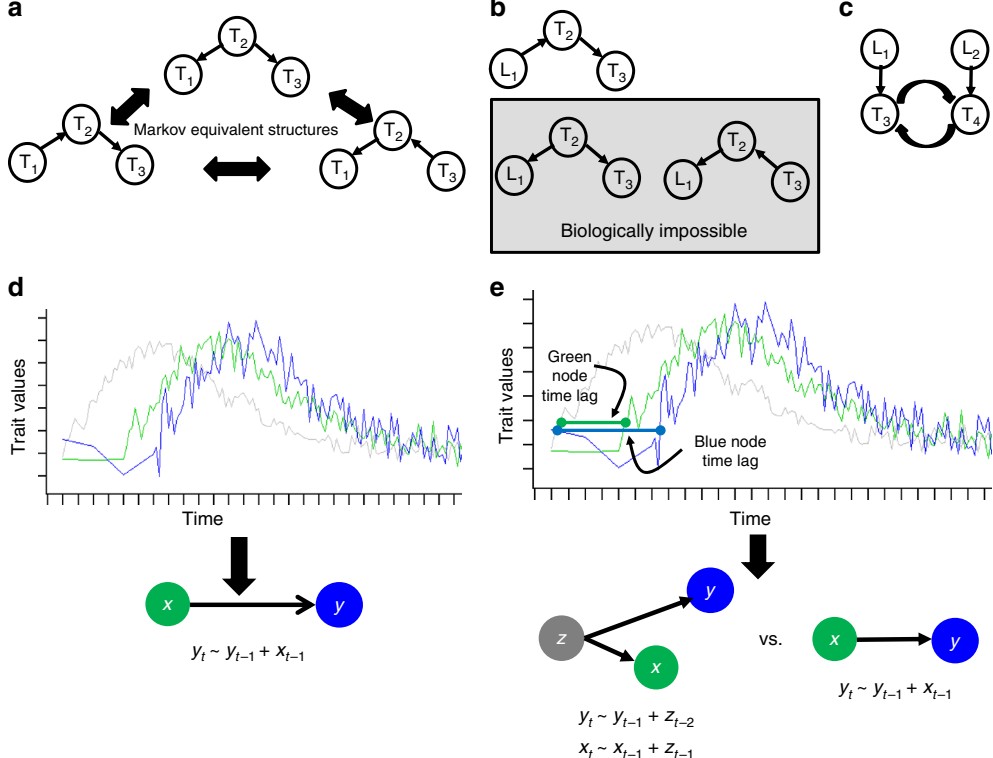

**Fig. 1** Overview of causal relationships in complex biological systems. **a** Bayesian network is a directed graph. However, causal relationships cannot be unambiguously inferred from a directed graph due to Markov equivalent structures. **b** Systematic genetic perturbation data enables causality inference and causal network construction by eliminating biologically impossible structures. **c** Two traits are related via a feedback loop. **d** Time-series data provide information for inferring causality. **e** Time-series data alone are not sufficient for distinguishing causality and varied time delay. Additional information is needed to infer true causal relationships

correlated, we develop MPTGA as a genetic association testing framework (see Methods). Instead of assuming the mean vectors of the multivariate normal distribution follow a logistic growth curve as in Ma et al.[25], we model the mean vectors of the expression trajectories using a polynomial function, which is able to capture diverse types of temporal responses.

Temporal QTL can be treated as a systematic source of perturbation to infer causality among traits associated with the QTL. There are a limited number of causal relationships possible between two traits associated with a given genetic locus[14,17] (Supplementary Fig. 1): simple causal/reactive models (M1 and M2), an independent model (M3), and partial causal/reactive models (M4 and M5). Based on these possible relationships, in the context of static QTL, a likelihood-based causality model selection (LCMS) procedure had been developed to infer causal relationships[14]. This approach has been widely validated as predicting causal relationships with reasonable accuracy[2,3,9,14,15,17]. In the context of multi-dimensional time-series data, we now seek to combine temporal and genetic information to infer causal relationships between two time series. Granger[26] formalized the idea of a time series-based causality test in the context of linear regression, where the prediction of a time series could be significantly improved by incorporating information from previous time points in a second time series. Several mediation models for longitudinal data were developed based on Granger causality[27], but no model takes genetic data into consideration. To develop a causality test based on genetics and time to assess how two traits are related, we adopted the idea of including the lagged values of the time series from one temporal-genetic associated trait to augment when comparing to the time series of the second temporal-genetic trait. More specifically, after

identifying two traits $X$ and $Y$ with temporal-genetic association to the same locus, there are five possible causal/reactive relationships as shown in Supplementary Fig. 1. In a causal model (M1: $X \rightarrow Y$), the genetic effect (or the association with the marker) of Trait $Y$ is solely explained by Trait $X$, so that the time-series values of Trait $Y$ can be predicted with values of Traits $X$ and $Y$ at previous time points. In an independent model (M3: $X \perp Y$), the genetic effect of Trait $Y$ cannot be explained by Trait $X$. In a partial causal model (M4), the genetic effect of Trait $Y$ can only partially be explained by Trait $X$, so that the time-series values of Trait $Y$ can be predicted with values of Traits $X$ and $Y$ at previous time points, as well as the genotype information at the associated locus. When traits $X$ and $Y$ were switched in models M1 and M4, the causal and partial causal relationships can be represented in models M2 and M5, respectively. First, we assess TGCT's power to distinguish causal/reactive relationships (M1 vs. M2) in general by comparing the joint likelihood $L(X, Y)$ (Methods). Then, we focus on the cis–trans trait pairs as the following: Trait $X$ has a *cis*-teQTL and Trait $Y$ has a trans-teQTL at the same locus so that models to be assessed are limited to M1, M3, and M4. We applied a linear regression on the corresponding time-series data for Trait Y and selected the model that best explains the data according to a given model selection criterion (e.g., Akaike information criterion (AIC) or Bayesian Information Criterion (BIC)) as detailed in Methods.

**Evaluating temporal-genetic association methods**. To compare the performance of multiple approaches for detecting temporal-genetic associations, we applied these methods to a set of simulated data (see Methods). Various time-series patterns were simulated (Supplementary Fig. 2), which were similar to the

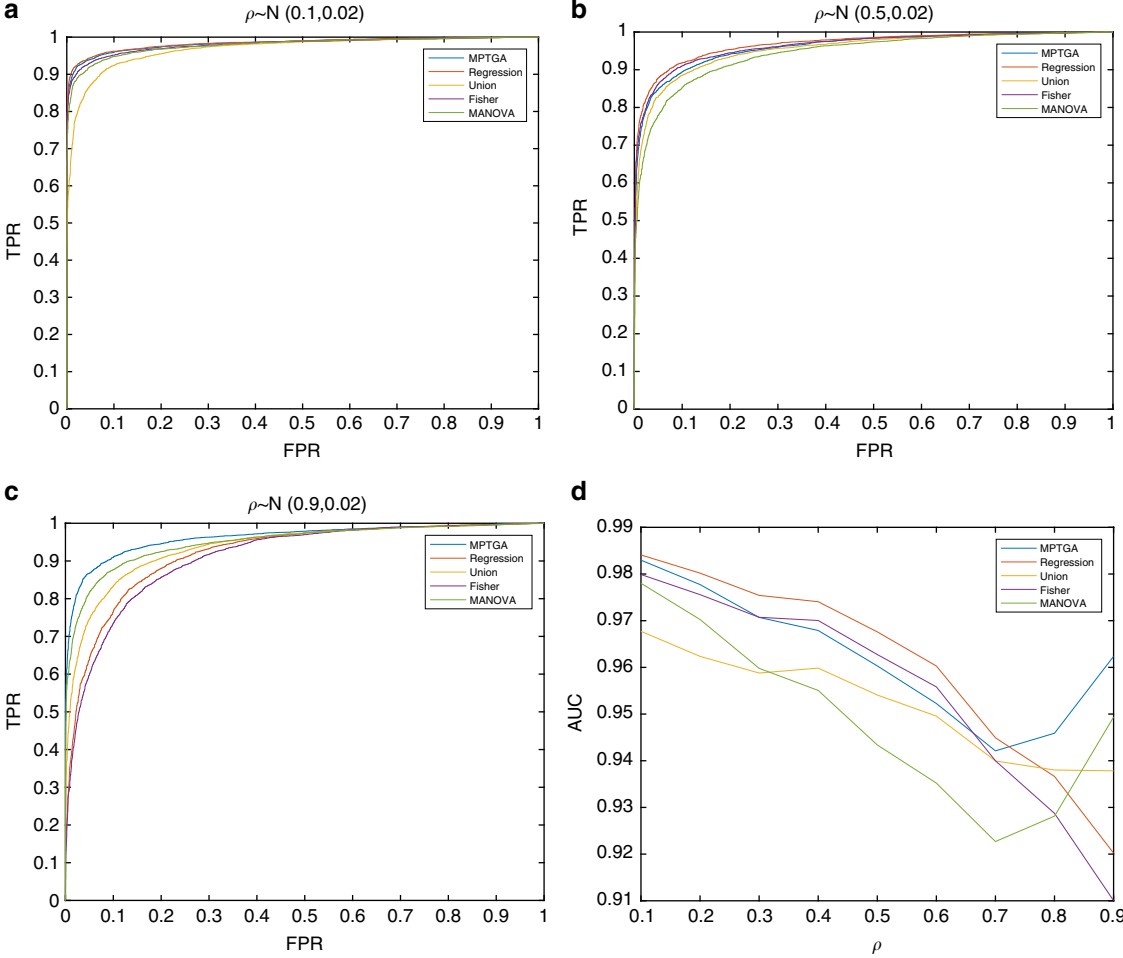

**Fig. 2** Performance comparison among different time-dependent genetic association methods. Performance was assessed under different strength of inter time point correlation (auto-correlation) based simulated data. **a** ROC curves of different methods when auto-correlation is low; **b** ROC curves of different methods when auto-correlation is intermediate; **c** ROC curves of different methods when auto-correlation is strong; **d** area under the curve (AUC) of different methods under different strength of auto-correlation

patterns observed in the yeast time-series data. As each time point in a time series is not independent, the residues at each time are correlated (auto-correlation). We generated time-series data assuming different strength of auto-correlation. Temporal genetic association results (Fig. 2) show that MPTGA performed the best in the context of strong auto-correlated data (MANOVA as the second best, the Fisher's method performed the worst), whereas MPTGA was essentially equivalent to the regression method in the context of weak auto-correlated data (the union method performed the worst). When the auto-correlation coefficient was around 0.7, all methods performed similarly, except MANOVA. The distribution of auto-correlation coefficients estimated from the empirical yeast time-series data was centered around 0.85 (Supplementary Fig. 3), at which the MPTGA performed best. In general, the MPTGA is robust over a broad range of operating conditions (Fig. 2d). The AR method was not included in Fig. 2, as it performed worse than other methods across all conditions (Supplementary Fig. 4). We also compared the power of these methods with different sample sizes, the pattern was similar as shown in Fig. 2 with MPTGA performing the best when auto-correlation was high (Supplementary Fig. 4). To assess a model's robustness when there are missing data (detailed in Methods), we randomly dropped data points in simulated time series at various rates and applied the above methods to the data sets with missing data. The performances of the MPTGA, Regression, and AR

methods were not sensitive to missing data, whereas the performances of the Union, Fisher, and MANOVA methods decreased as the missing data rate increased (Supplementary Fig. 5).

**Evaluating the TGCT test**. To evaluate the performance of TGCT, we simulated pairs of time-series data according to causal, independent, or partial causal models (see Methods). We applied TGCT only to the pairs in which both time-series traits $X$ and $Y$ had temporal-genetic associations (MPTGA $p < 10^{-6}$) to the tested locus. First, we simulated pairs of traits according to the causal model (M1), then assessed whether the causal (M1: $X \rightarrow Y$) or reactive (M2: $Y \rightarrow X$) model fit the data better (Methods). TGCT identified the correct model in most cases with the accuracy of 99.54%, 99.82%, 99.95%, and 99.97% for the sample size of 20, 50, 100, 150, respectively (Supplementary Fig. 6, the log likelihood ratio (LR) of M1 vs. M2 is shown in Supplementary Fig. 7). When genetic information is known, we can focus on relationships between cis-regulated genes and trans-regulated genes instead of testing all possible pairs. In the rest of the tests, we assumed Trait $X$ had a cis-teQTL so that we can simplify our tests without considering models M2 and M5. When comparing models M1, M3, and M4, we needed to model only Trait $Y$ without explicitly modeling Trait $X$ (Methods). For pairs simulated under the causal model (M1), TGCT identified the causal model as the best model in all cases across a wide range of

**Table 1 Number of eQTL identified and average 1-LOD drop CI by different approaches (static, MPTGA, union, regression, and Fisher's method)**

|                     | Static (T0) | MPTGA | Union | Regression | Fisher |
|---------------------|-------------|-------|-------|------------|--------|
| #eQTLs identified   | 3288        | 3819  | 3669  | 3453       | 3011   |
| Avg. 95% CI (kb)    | 37.5        | 28.7  | 30.8  | 17.9       | 35     |

CI confidence interval, MPTGA multivariate polynomial temporal genetic association

strength of AR and causal effects (Supplementary Fig. 8). The BIC differences between the causal model and other models are shown in Supplementary Fig. 9. For pairs simulated under the independent model (M3), TGCT identified the independent model as the best model in most cases with accuracy of 95.8%, 97.7%, 97.7%, and 98.9% for the sample size of 20, 50, 100, and 150, respectively, across a wide range of parameters (Supplementary Fig. 10). Simulations under the partial causal model (M4) were complicated as there were three parameters for representing the strength of genetic and causal effects (Supplementary Fig. 11). TGCT identified the partial causal model as the best model in all cases except when the genetic or causal effect was close to zero. For example, when $\beta_2$ is close to 0, the partial model (M4) is converted to the independent model (M3). In such cases, TGCT identified the independent model as the best model. When both $\beta_{10}$ and $\beta_{11}$ are close to 0, the partial model (M4) is converted to causal model (M1) and TGCT identified the causal model as the best model in such cases.

**Dissecting regulatory networks response to rapamycin**. We applied multiple methods to expression data generated in a population of 95 genotyped haploid yeast segregants that were treated with the macrolide drug rapamycin[28] and compared teQTLs identified at a 5% false discovery rate (FDR) (teQTLs are listed in Supplementary Tables 1–5). The yeast segregants were profiled at six different time points starting with a baseline expression profile just before treatment and then five subsequent time points post treatment. The aim in applying the teQTL and causality analysis in this population was to dissect the causal regulators most strongly modulating the treatment response across individuals in the population. Given traditional eQTL detection methods considering gene expression levels in a static state (without considering the time-series data), for a baseline to use in the comparisons, we mapped eQTLs based on gene expression data at the first time point.

Compared with the teQTL approaches, the static eQTL approach detected fewer QTLs (Table 1) at a fixed FDR. Among the four teQTL methods MPTGA, union, regression, and the Fisher's *p*-value methods resulted in the highest to lowest numbers of teQTLs, respectively. When comparing teQTL confidence intervals for constraining the true location of variant (s) underlying the teQTL, the QTL 95% confidence intervals (Methods) for all teQTL methods were tighter compared with the static methods (Table 1). These results suggest that the methods that take into account the time-series data refine the QTL location, thus reducing the number of potential candidate causal regulators to consider in the linkage region. The regression method resulted in fewer but sharper eQTLs than the MPTGA method. MPTGA is the best model with balance of the number of eQTLs identified and average confidence intervals.

Similar to the eQTLs in this yeast cross that have been previously reported[3,15,29,30], the teQTLs were clustered into teQTL hotspots (Fig. 3). Across all methods a total of 18 hotspots were identified (Table 2, expression traits linked to each eQTL hotspot are listed in Supplementary Tables 6–10). Of the 14 eQTL hotspots identified by the static eQTL method, 10 overlapped

with eQTL hotspots previously identified in this same yeast F2 cross[3,29,30]. Among the ten eQTL hotspots, seven of them identified by the static were identified by all four times series based methods. In addition, three and three additional teQTL hotspots were identified by the MPTGA and regression methods, respectively (Table 2). Despite rapamycin treatment inducing a large impact on cell cycle and metabolism in yeast[31], none of the 14 static eQTL hotspots were enriched for genes in the rapamycin transcriptional response signature[31]. Of the 11 teQTL hotspots identified by MPTGA, eight were significantly enriched for this signature. A key mechanism for cell growth is the regulation of ribosome biogenesis. Ribosomal protein gene expression is regulated by mTOR, the target of rapamycin[32]. Six of the eight teQTL hotspots identified by MPTGA enriched for the rapamycin response signature were also enriched for the GO term structural constituent of ribosome (Table 3), demonstrating the ability of MPTGA to capture both static and dynamic genetic associations.

In contrast, only one (chrXV:150,000) of the teQTL hotspots identified by the union method were enriched for the rapamycin response signature, suggesting that although this approach increases the power to detect static eQTL, the union method is not as sensitive for detecting a dynamic response. The regression method is closely related to the MPTGA method, but only one (chrXV:150,000) of its identified hotspots was enriched for genes in the rapamycin response signature, suggesting this approach in a temporal context may be prone to sporadic associations.

**Inferring causal regulators of teQTL hotspots**. Similar as we have previously shown for static eQTL hotspots[3,15,29], we applied TGCT to resolve the causal regulators underlying teQTL hotspots. For each teQTL hotspot, candidate causal genes were defined as genes with cis-teQTLs linked to the teQTL hotspot. We applied TGCT to infer the causal regulators of the teQTL hotspots identified by MPTGA (Table 4, all predicted causal relationships are listed in Supplementary Table 11). The distribution of BIC difference between causal model M1 and the second best fit models is shown in Supplementary Fig. 12. The top putative causal regulators for each teQTL hotspot were ranked based on the number of causal relationships that the regulator had. We compared the causal regulators identified in this dataset to those we previously predicted and validated[3,29,30].

Of the 11 teQTL hotspots identified by MPTGA, 7 overlapped previously identified static eQTL hotspots in this population (Table 2) for which we had predicted and validated static causal regulators[3,29]. All previously validated causal regulators were identified as putative key causal regulators by TGCT (bolded genes in Table 4). In addition to the eQTL hotspots identified by the static method, MPTGA identified 3 dynamic teQTL hotspots. The teQTL hotspot at chrV:190,000 was the largest in terms of the number of expression traits linking to the locus, with 162 gene expression traits linked to this locus versus only 7 expression traits identified by the static eQTL approach. The genes linked to this teQTL hotspot were significantly enriched for the rapamycin response signature (3.9-fold enrichment; Fisher's exact test(FET) $p = 1.28 \times 10^{-10}$). The top putative causal regulator predicted by TGCT for this hotspot was *ISC1*, inositol phosphingolipid

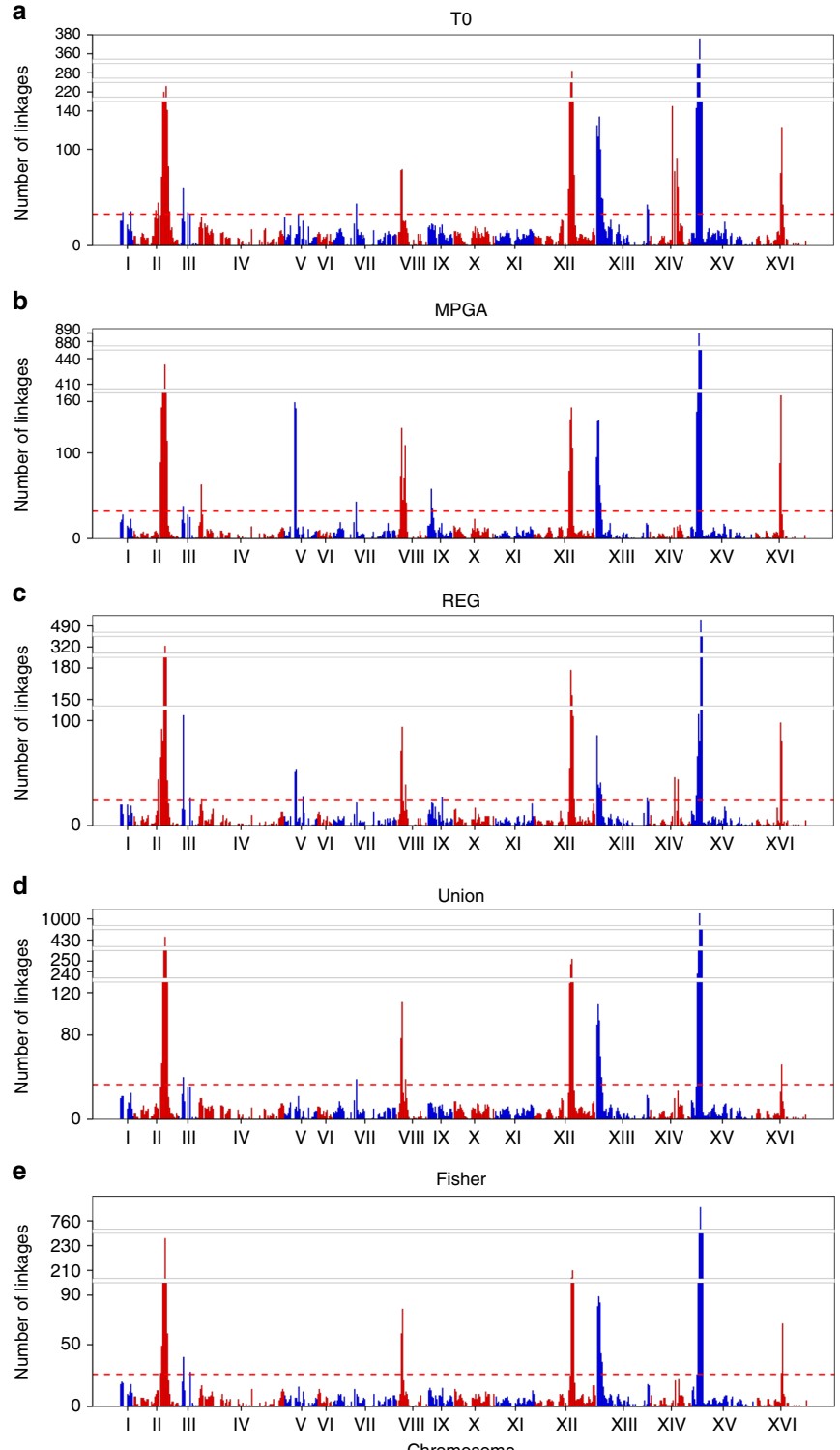

**Fig. 3** Number of eQTLs linked to each genome location based on different approaches. The genome was divided into 602 bins of 20 kb each and shown in chromosomal order. The red lines were the cutoff for eQTL hotspot identification. **a** The static method; **b** the MPTGA method; **c** the regression method; **d** the union method; **e** the Fisher's *p*-value method

phospholipase C, a gene involved in ceramide production[33]. *ISC1* was supported as causal for 136 of the 162 genes linked to this teQTL hotspot. Rapamycin induces insulin resistance via mTORC2[34], which regulates de novo ceramide synthesis[35]. Ceramide and its metabolites also play a pathogenic role in insulin resistance[36]. Taken together, these data support *ISC1* as a causal regulator for rapamycin response differences among the yeast segregants. The identification of this teQTL hotspot and of *ISC1* as a causal regulator could not have happened by analyzing any single time point after the rapamycin treatment. The genetic-by-drug perturbation interaction at this locus was only detectable in light of the time-series data considered in full.

**Table 2 Hotspots identified by different approaches and enrichment analysis of rapamycin response signature**

| Hotspot | Chr | Pos | Size of eQTL hotspot | | | | | Enrichment for the rapamycin signature | | | | | | | | | |
|---|---|---|---|---|---|---|---|---|---|---|---|---|---|---|---|---|---|
| | | | T0 | MPTGA | UNION | Regre. | Fisher | T0 | | MPTGA | | UNION | | Regression | | Fisher | |
| | | | | | | | | perc (%) | p-value | perc (%) | p-value | perc (%) | p-value | perc (%) | p-value | perc (%) | p-value |
| 1 | 1 | 50,000 | 34[a] | NA | NA | NA | NA | 2.9 | 1 | NA | NA | NA | NA | NA | NA | NA | NA |
| 2 | 1 | 190,000 | 35 | NA | NA | NA | NA | 0 | 1 | NA | NA | NA | NA | NA | NA | NA | NA |
| 3 | 2 | 430,000 | 44[a] | NA | NA | 44 | NA | 0 | 1 | NA | NA | NA | NA | 5.9 | 0.61 | NA | NA |
| 4 | 2 | 550,000 | 367[a] | 662 | 584 | 607 | 372 | 5.4 | 0.24 | 7.7 | $9.59e{-}5$[b] | 4.8 | 0.42 | 5.9 | 0.06 | 5.9 | 0.13 |
| 5 | 3 | 90,000 | 60[a] | 38 | 40 | 105 | 40 | 10 | 0.0551 | 18.4 | $1.45e{-}3$[b] | 12.5 | 0.034 | 8.6 | 0.05 | 12.5 | 0.034 |
| 6 | 3 | 170,000 | 34 | NA | NA | 26 | 28 | 8.8 | 0.2022 | NA | NA | NA | NA | 11.5 | 0.11 | 10.7 | 0.13 |
| 7 | 4 | 90,000 | NA | 63[c] | NA | 25[c] | NA | NA | NA | 20.6 | $3.85e{-}6$[b] | NA | NA | 4.0 | 1 | NA | NA |
| 8 | 5 | 190,000 | NA | 162[c] | NA | 57[c] | NA | NA | NA | 17.9 | $1.28e{-}10$[b] | NA | NA | 8.8 | 0.12 | NA | NA |
| 9 | 7 | 410,000 | 43 | 43 | 38 | NA | NA | 2.3 | 1 | 0 | 1 | NA | NA | 0.0 | 1 | NA | NA |
| 10 | 8 | 110,000 | 97[a] | 256 | 121 | 111 | 87 | 6.2 | 0.28 | 4.3 | 0.63 | 4.1 | 0.66 | 2.7 | 0.89 | 3.4 | 0.77 |
| 11 | 9 | 70,000 | NA | 65[c] | NA | NA | NA | NA | NA | 23.1 | $1.44e{-}7$[b] | NA | NA | NA | NA | NA | NA |
| 12 | 9 | 250,000 | NA | NA | NA | 27[c] | NA | NA | NA | NA | NA | NA | NA | 3.7 | 1 | NA | NA |
| 13 | 12 | 670,000 | 363[a] | 171 | 295 | 267 | 248 | 2.8 | 0.97 | 4.1 | 0.67 | 2.7 | 0.97 | 2.2 | 0.99 | 2.8 | 0.94 |
| 14 | 13 | 70,000 | 239[a] | 237 | 201 | 163 | 172 | 7.1 | 0.046 | 9.7 | $4.64e{-}4$[b] | 6.0 | 0.21 | 5.5 | 0.33 | 7.0 | 0.095 |
| 15 | 13 | 910,000 | 42 | NA | NA | 26 | NA | 4.8 | 0.58 | NA | NA | NA | NA | 7.7 | 0.34 | NA | NA |
| 16 | 14 | 410,000 | 145[a] | NA | NA | 46 | NA | 2.1 | 0.97 | NA | NA | NA | NA | 6.5 | 0.35 | NA | NA |
| 17 | 15 | 150,000 | 486[a] | 1000 | 1183 | 742 | 987 | 6.8 | 0.0131 | 7.7 | $7.71e{-}7$[b] | 6.1 | 0.0042[b] | 6.7 | 0.0026[b] | 6.3 | 0.0041[b] |
| 18 | 16 | 510,000 | 138[a] | 168 | 52 | 147 | 67 | 3.6 | 0.76 | 11.9 | $7.08e{-}5$[b] | 7.7 | 0.21 | 4.8 | 0.51 | 4.4 | 0.6 |

An NA in the table means the locus was not an eQTL hotspot in the corresponding method. When comparing with the rapamycin response signature, the percentage of genes in a hotspot overlapping with the rapamycin signature and the Fisher's exact test p-value are reported. The expected percentage by chance is 4.6%. The hotspots significantly enriched for the rapamycin signature were marked ($p < 0.05$ after multiple testing correction)
[a]Hotspots overlapped with those previously defined static eQTL hotspots
[b]The hotspots significantly enriched for the rapamycin signature ($p < 0.05$ after multiple testing correction)
[c]The hotspots identified only via time-dependent approaches
eQTL expression quantitative trait loci, MPTGA multivariate polynomial temporal genetic association

**Table 3 GO term enrichment analysis for eQTL hotspots identified by MPTGA**

| Hotspot | | GO-Term | p-value |
|---|---|---|---|
| Chr | Pos | | |
| 2 | 550,000[a] | Structural constituent of ribosome | 3.97E−57 |
| 3 | 90,000[a] | Branched chain family amino acid | 2.08E−10 |
| 4 | 90,000[a] | Structural constituent of ribosome | 2.03E−37 |
| 5 | 190,000[a] | Structural constituent of ribosome | 1.27E−126 |
| 7 | 410,000 | Protein folding | 6.85E−14 |
| 8 | 110,000 | Mating projection tip | 7.27E−6 |
| 9 | 70,000[a] | Structural constituent of ribosome | 1.77E−50 |
| 12 | 670,000 | Ergosterol biosynthesis | 6.37E−25 |
| 13 | 70,000[a] | Phosphate transport | 2.52E−5 |
| 15 | 150,000[a] | Structural constituent of ribosome | 7.25E−43 |
| 16 | 510,000[a] | Structural constituent of ribosome | 4.00E−28 |

[a]The hotspots were enriched for the rapamycin signature as in Table 2
eQTL expression quantitative trait loci, MPTGA multivariate polynomial temporal genetic association

In addition to the *ISC1* teQTL hotspot, the teQTL hotspot at locus chrIX:70,000 was only identified by the MPTGA method. A general temporal pattern of genes linked to the hotspot is shown in Fig. 4. The gene expression level differences between segregants carrying different genotype at the locus were small but consistently getting larger (Fig. 4a, Supplementary Fig. 14). Testing each time point individually, the differences were not significant, which may explain why the static, union, and Fisher's method could not identify the teQTL hotspot. MPTGA, the regression, and the union methods (Fig. 4b–d) suggested a putative teQTL at the locus, but p-values for the regression and the union methods were not significant at an FDR < 0.05. Without constraining on residues, the regression method is prone to sporadic association[37] (detailed in Discussion) so that p-value cutoff for 5% FDR is much lower than the one for the MPTGA, which explained why the regression method missed the teQTL hotspot.

The chrIX:70,000 teQTL hotspot was also significantly enriched for rapamycin signature genes (5.0-fold enrichment, FET $p = 1.4 \times 10^{-7}$), which suggests this teQTL hotspot was driven by gene-by-rapamycin interactions. The top putative causal regulator predicted by TGCT for this hotspot was *RRD1* (Table 4, the distribution of BIC difference between causal model M1 and the second best fit models is shown in Supplementary Fig. 13). *RRD1* is an activator of *PP2A*, a gene involved in G1 phase progression. *PP2A* is required for rapamycin response[38], directly supporting our prediction that *RRD1* mediates rapamycin response variation among the yeast segregants. To validate *RRD1* as a causal regulator of the teQTL hotspot at chrIX:70,000 driven by gene-by-rapamycin interactions, we compared genome-wide gene expression profiles of the *RRD1* knockout strain to the wild-type strain, both with and without rapamycin in the culture media (Methods). At an FDR < 1%, 64 differentially expressed genes (DEGs) were identified between the *RRD1* knockout and

**Table 4 Causal regulators for teQTL hotspots inferred by the TCGT test**

| Hotspot | | Yvert et al. | BN full | TGCT |
|---|---|---|---|---|
| Chr | Pos | | | |
| 2 | 550,000[a] | ***AMN1***, *MAK5* | *TBS1, TOS1,ARA1, CSH1, SUP45, CNS1,* ***AMN1*** | *ICS2, RPB5, UBS1, TYR1, YSW1, SLI15,* ***AMN1***, *TBS1, TOS1* |
| 3 | 90,000[a] | ***LEU2*** | ***LEU2***, *ILV6, NFS1, CIT2, MATALPHA1* | ***LEU2***, *FRM2, YCL021W-A* |
| 4 | 90,000[a] | NA | NA | *ASF2, YDL203C, PRR2* |
| 5 | 190,000[a] | NA | NA | *ISC1, SPC25, GPA2* |
| 7 | 410,000 | NA | NA | *MST27, TIF4632* |
| 8 | 110,000 | ***GPA1*** | ***GPA1*** | *ERG11, LAG1, YHL008C, SHU1,* ***GPA1*** |
| 9 | 70,000[a] | NA | NA | *RRD1, ECM37, YIL151C, CCT2* |
| 12 | 670,000 | ***HAP1*** | ***HAP1*** | *DCS1,* ***HAP1***, *PIG1, LCB5, PDR8* |
| 13 | 70,000[a] | None | None | *MDM1, RPM2,TAF13, COQ5* |
| 15 | 150,000[a] | None | ***PHM7*** | *MSH2, PKH2, HAL9, RFC4, YOL098C, COQ3,* ***PHM7*** |
| 16 | 510,000[a] | NA | NA | *MET12, RMI1, PHO85* |

The putative causal regulators were compared with previous predictions in Yvert et al.[29] and Zhu et al.[3]
Genes in bold font are overlapping with previous findings
[a]The hotspots were enriched for the rapamycin signature as in Table 2
*NA* not applicable, *teQTL* temporal expression quantitative trait loci, *TGCT* temporal genetic causality test

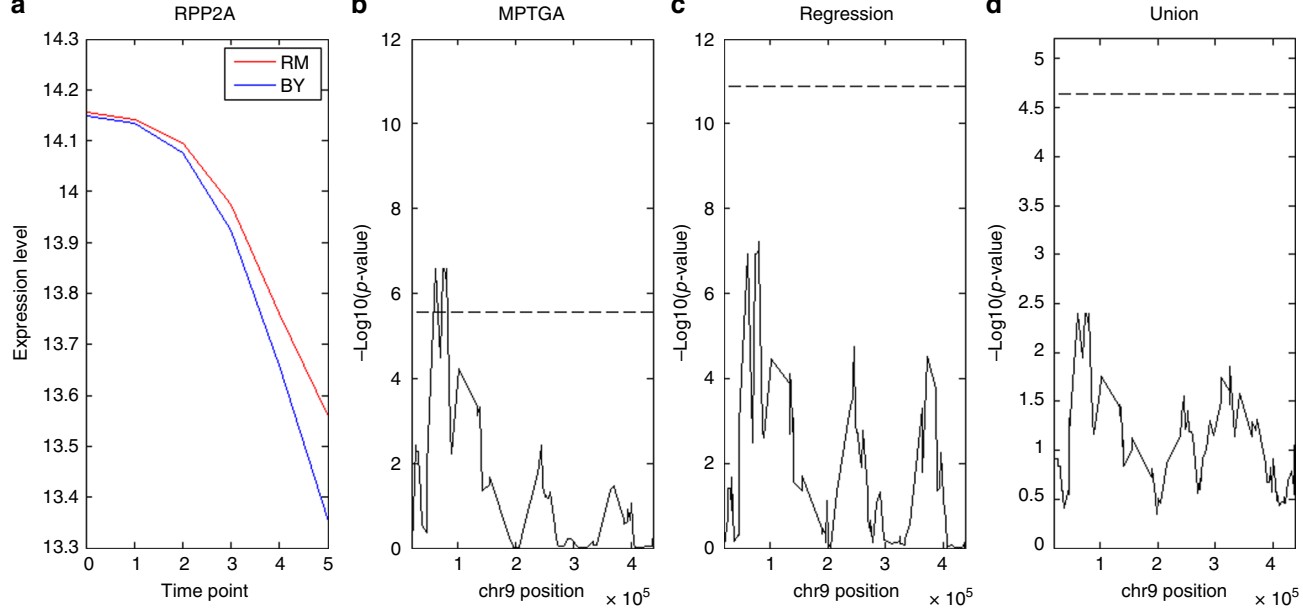

**Fig. 4** The teQTL hotspot chrIX:70,000 was only identified by the MPTGA method. **a** A typical pattern of expression traits linked to the teQTL hotspot. **b** Plot of − log 10(*p*-value) of *RPP2A* association at chrIX based on the MPTGA method. **c** Plot of − log 10(*p*-value) of *RPP2A* association at chrIX based on the regression method. **d** Plot of − log 10(*p*-value) of *RPP2A* association at chrIX based on the union method. Dashed lines in (**b**, **c**), and **d** are the *p*-value cutoff corresponding to FDR = 0.05

wild-type strains, without exposure to rapamycin. These 64 DGEs were significantly overlapped with the rapamycin signature (5.1-fold enrichment, FET $p = 1.1 \times 10^{-7}$). When compared with genes linked to the 11 teQTL hotspots identified by MPTGA, the *RRD1* knockout signature significantly overlapped with 5 teQTL hotspots (Fig. 5a), which were also enriched for the rapamycin signature (Table 2). The teQTL hotspot ChrIX:70,000 was enriched for the *RRD1* knockout signature (4.1-fold enrichment, FET $p = 0.036$) and the teQTL hotspot ChrXV:150,000 was most significantly enriched for the *RRD1* knockout signature (2.5-fold enrichment, FET $p = 8.3 \times 10^{-7}$). When compared with the eQTL hotspots based on static T0 data, the *RRD1* knockout signature significantly overlapped with the eQTL hotspot at ChrXV:150,000 (4.2-fold enrichment, FET $p = 8.1 \times 10^{-10}$). The *RRD1* knockout signature was enriched for the GO biological process response to stress (12.9-fold enrichment, FET $p = 3.1 \times 10^{-9}$), which is consistent with the functional annotation of this

static eQTL hotspot[3,29]. These results were consistent with *RRD1* expression levels being regulated both in cis and in trans by DNA variations at ChrXV:150,000 (Fig. 5b). When comparing the *RRD1* knockout and wild-type strains in the presence of rapamycin, 582 DGEs were identified at an FDR < 1%. The *RRD1* rapamycin signature overlapped seven teQTL hotspots (Fig. 5a), which were all enriched for the rapamycin signature. Among these teQTL hotspots, the teQTL hotspot ChrIX:70,000, where *RRD1* is physically located, was with the highest fold enrichment (7.1-fold enrichment, FET $p = 3.7 \times 10^{-33}$). More specifically, directions of changes for all genes in the overlap between genes linked to the teQTL hotspot ChrIX:70,000 and DGEs in *RRD1* knockout signature are consistent between the time course and *RRD1* knockout experiments. The segregants carrying RM allele at the *RRD1* locus had low *RRD1* expression level in comparison with the segregants carrying BY allele (Supplementary Fig. 15). Among 65 genes linked to the teQTL

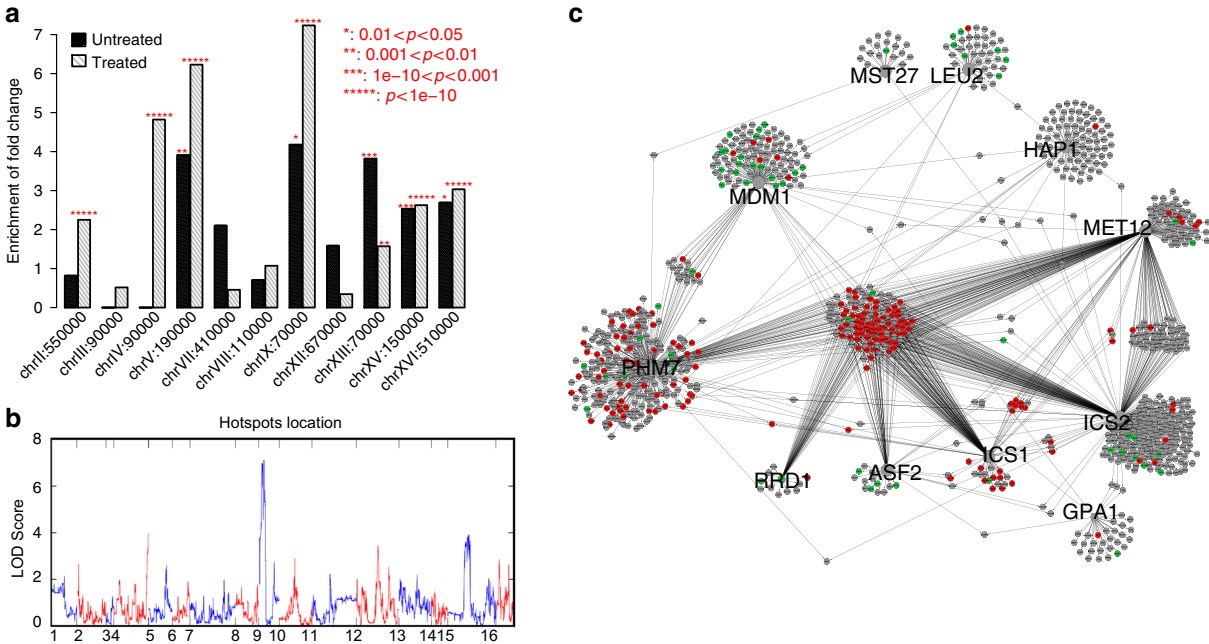

**Fig. 5** *RRD1* is a putative key causal regulator for the teQTL hotspot chrIX:70,000. **a** *RRD1* ko signatures without and with rapamycin treatment were only compared with teQTL hotspots identified by MPTGA. *RRD1* ko signature with rapamycin treatment (*RRD1* X rapamycin interaction signature) is enriched in multiple teQTL hotspots including ones at chrIX:70,000 and chrXV:170,000. **b** The teQTL plot for *RRD1* based on MPTGA. *RRD1* physically locates at chrIX:70,000 and has a strong cis-eQTL at the locus and a trans-eQTL at chrXV:150,000. **c** The causal network based on TGCT indicates that genes linked to the teQTL hotspot chrIX:70,000 are also regulated by multiple genetic perturbations so that *RRD1* ko signature with rapamycin treatment overlaps with multiple teQTL hotspots. Nodes of larger size are putative key causal genes for teQTL hotspots. Red and green nodes represent upregulated and downregulated genes in *RRD1* ko, in comparison to wild-type strain when treated with rapamycin, respectively

hotspot chrIX:70,000, 56 genes were expressed higher in the segregants carrying RM allele and 42 of them overlapped with upregulated in *RRD1*-KO strain (FET $p = 2.4 \times 10^{-39}$), whereas 9 genes were expressed lower in the segregants carrying RM allele and 6 of them overlapped with downregulated genes in *RRD1*-KO strain (FET $p = 2.0 \times 10^{-7}$). In addition to the hotspot ChrIX:70,000, the top three teQTL hotspots with the highest fold enrichment include ChrV:190,000 and ChrIV:90,000 (fold enrichment = 6.2 and 4.7, FET $p = 1.2 \times 10^{-24}$ and $2.9 \times 10^{-14}$, respectively), which are the three unique teQTL hotspots compared with static eQTL hotspots (Table 2). The teQTL hotspot at chrXV:150,000 was with the most significant enrichment *p*-value (2.6-fold enrichment, FET $p = 1.3 \times 10^{-64}$), consistent with *RRD1* expression variation being linked in cis to ChrIX:70,000 and in trans to ChrXV:150,000 (Fig. 5b). Genes putatively regulated by *RRD1* inferred by TGCT were also regulated by other genetic perturbations (Fig. 5c). The *RRD1* rapamycin signature was significantly enriched for the GO term structural constituent of ribosome (4.0-fold enrichment, FET $p = 1.6 \times 10^{-33}$), which is consistent with the GO functional annotations of the set of genes simultaneously linked to these teQTL hotspots (Table 3). These results combined indicate that *RRD1* interacts with rapamycin to give rise to the teQTL hotspot at chrIX:70,000, and that this gene-by-rapamycin interaction was only detected by our TGCT test.

## Discussion

In this study, we developed MPTGA to optimally integrate genetic and temporal information to identify genetic associations for gene expression traits. With respect to other methods we tested, MPTGA was the most robust and sensitive in our simulation study. When applied to a yeast F2 time-series data set profiled in response to a rapamycin perturbation, MPTGA

detected more biologically relevant teQTL hotspots, along with tighter eQTL confidence intervals compared with the static method (Table 1), which may lead to fewer candidate causal regulators to consider for each eQTL hotspot. We also developed the causal inference test, TGCT, which simultaneously considers temporal and genetic data to infer causal relationships systematically. Temporal-genetic data together has more power to distinguish which gene is the true causal regulator among correlated genes colocalizing at a locus than the static method. Application of TGCT in the F2 yeast cross in the context of treatment with rapamycin resulted in the identification the key causal regulators *ISC1* and *RRD1*, which modulated response to this perturbation, revealing the molecular mechanisms related to rapamycin response. Our prediction of *RRD1* as a causal regulator of gene-by-rapamycin interactions was experimentally confirmed.

For each teQTL hotspot, we tested all cis–trans gene pairs at the locus for potential causal relationships. For a gene with a trans-eQTL at a hotspot, the TGCT may report multiple candidate causal genes. On the other hand, for two genes ($X1$ and $X2$) with cis-eQTLs at a hotspot, both may be causal to an overlapped set of genes ($Ys$) with trans-eQTLs linked the locus, e.g., $X1 \rightarrow Y$ and $X2 \rightarrow Y$. In these cases, TGCT cannot distinguish which cis-eQTL gene is the true causal gene. Thus, multiple putative causal genes were reported for hotspots with a large number eQTLs (Table 4). To distinguish which causal relationship, $X1 \rightarrow Y$ or $X2 \rightarrow Y$, is true, more data are needed, such as more F2 strains to breakdown linkage disequilibrium (LD) structures or more time points to break correlation relationships among colocalized genes so that the TGCT has more power to distinguish which gene is true causal regulator among correlated genes colocalizing at a locus than static methods. Follow-up experiments are recommended to validate putative causal regulators.

Rapamycin has been shown to extend lifespan in mice[39], but then chronic usage has also been shown to lead to insulin

**Table 5 Enrichment analysis of genes that can extend yeast replicative lifespan in hotspots**

| Hotspot | | T0 | | MPTGA | | UNION | | REGRESSION | | FISHER | |
|---|---|---|---|---|---|---|---|---|---|---|---|
| CHR | Pos | Fold enrichment | p-value | Fold enrichment | p-value | Fold enrichment | p-value | Fold Enrichment | p-value | Fold Enrichment | p-value |
| 1 | 50,000 | 0[a] | 1 | NA | NA | NA | NA | NA | NA | NA | NA |
| 1 | 190,000 | 1.37 | 0.43 | NA | NA | NA | NA | NA | NA | NA | NA |
| 2 | 430,000 | 0[a] | 1 | NA | NA | NA | NA | 1.63 | 0.28 | NA | NA |
| 2 | 550,000 | 0.72[a] | 0.91 | 1.77 | 3.27E−5[b] | 0.98 | 0.57 | 1.14 | 0.24 | 0.97 | 0.6 |
| 3 | 90,000 | 0.4[a] | 1 | 0 | 1 | 0.6 | 1 | 0.68 | 0.82 | 0 | 1 |
| 3 | 170,000 | 0 | 1 | NA | NA | NA | NA | 0 | 1 | 0 | 1 |
| 4 | 90,000 | NA | NA | 3.04[c] | 0.004[b] | NA | NA | 0[c] | 1 | NA | NA |
| 5 | 190,000 | NA | NA | 4.29[c] | 1.29E−11[b] | NA | NA | 5.04[c] | 2.81E−6[b] | NA | NA |
| 5 | 330,000 | NA | NA | NA | NA | NA | NA | 0.86 | 1 | NA | NA |
| 7 | 410,000 | 2.79 | 0.03 | 1.67 | 0.27 | 1.89 | 0.21 | NA | NA | NA | NA |
| 8 | 110,000 | 0[a] | 1 | 0.47 | 0.98 | 0.2 | 1 | 0.22 | 1 | 0.28 | 1 |
| 9 | 70,000 | NA | NA | 4.42[c] | 1.20E−5[b] | NA | NA | NA | NA | NA | NA |
| 9 | 250,000 | NA | NA | NA | NA | NA | NA | 0.89[c] | 1 | NA | NA |
| 12 | 670,000 | 0.99[a] | 0.55 | 0.98 | 0.58 | 1.14 | 0.35 | 1.17 | 0.32 | 1.35 | 0.15 |
| 13 | 70,000 | 0.5[a] | 0.97 | 0.61 | 0.94 | 0.83 | 0.74 | 0.88 | 0.68 | 0.84 | 0.73 |
| 13 | 910,000 | 2.28 | 0.1 | NA | NA | NA | NA | 3.69 | 0.02 | NA | NA |
| 14 | 490,000 | 1.13[a] | 0.42 | NA | NA | NA | NA | 1.56 | 0.3 | NA | NA |
| 15 | 150,000 | 0.89[a] | 0.74 | 1.25 | 0.05 | 0.87 | 0.87 | 0.87 | 0.81 | 0.83 | 0.91 |
| 16 | 510,000 | 1.39[a] | 0.22 | 2 | 0.01 | 0.92 | 0.65 | 0.98 | 0.58 | 0.72 | 0.78 |

Similar to Table 2
[a]Hotspots are overlapped with those previously defined static eQTL hotspots
[b]The hotspots were enriched for the rapamycin signature
[c]The hotspots were identified only via time-dependent approaches
*MPTGA* multivariate polynomial temporal genetic association, *NA* not applicable

resistance[34]. Identifying what molecular and physiological states stand to benefit from rapamycin treatment is critical before such a drug can be considered as an anti-aging treatment. A systematic screen identifies 238 genes whose deletion extends replicative lifespan in yeast[40]. The 238 aging related gene set marginally overlapped with the rapamycin signature[31] (1.7-fold enrichment, FET $p = 0.03$). However, the three unique teQTL hotspots identified by the MPTGA were significantly enriched for the aging related genes (fold enrichment = 3.0, 4.3, and 4.4, FET $p = 0.004$, $1.3 \times 10^{-11}$, and $1.2 \times 10^{-5}$, for ChrIV:90,000, ChrV:190,000, and ChrIX:70,000, respectively), whereas none of the static eQTL hotspots nor eQTL hotspots identified by the union nor the Fisher's method was enriched for the aging genes at $p < 0.01$ (Table 5). As we show in Fig. 3 and Table 2, gene-by-rapamycin interactions were only detected when a time series was considered as a whole, not when individual time points were considered separately, directly demonstrating the importance of a temporal genetic association study. Most gene-by-perturbation interaction screens, such as synthetic lethal small interfering RNA screen[41], monitor effects at only one time point. Our results suggest that monitoring such effects at multiple time points and analyzing them together as a time series can dramatically increase the power of detecting gene-by-perturbation interactions, as well as causal relationships among traits.

MPTGA and the regression methods are closely related (Methods). Even though the p-values of the two methods are not directly comparable (as the underlying functions for modeling are different), the association p-values based on the two methods were closely tracked with each other (an example in Fig. 4b, c; Supplementary Fig. 16) in general. However, there were multiple differences in the yeast teQTL results. Given variance observed in a given variable of interest, the regression model attempts to fit as much of the variance as possible so that it is more prone to sporadic associations or overfitting[37]. In contrast, MPTGA is a regularized regression method, which is constrained by regularization terms and so can only fit a portion of the variance. To access the tendency of sporadic association or overfitting in each method, we compared p-values of neighboring single-nucleotide polymorphisms (SNPs) (Methods). The p-values for peak SNPs

and neighboring SNPs were highly correlated, with correlation coefficients of 0.89 and 0.69 for MPTGA and the regression method, respectively (Supplementary Fig. 17, detailed in Supplementary Discussion), suggesting that MPTGA is less prone to overfitting than the regression method.

We also checked statistical validity and potential inflation of p-values of MPTGA (detailed in Methods). First, we simulated a set of gene expression traits and genotypes. As they were simulated independently, no gene association was expected. The QQ plot for the simulated data (Supplementary Fig. 18a) suggests that p-values of MPTGA are slightly inflated. We then generated permuted data from the yeast F2 data set by permuting strain labels so that the genetic structure is intact. As there are LD structures in the genetic structure, the QQ plot for the permuted data (Supplementary Fig. 18b) is slightly different from the QQ plot for the simulated data. The QQ plot for the yeast F2 data (Supplementary Fig. 18c) is significantly different from the plots for simulated and permuted data. The QQ plot comparing the results of the real data and permuted data (Supplementary Fig. 18d) indicates that the result of the real data was significantly different from the result of the permutated data. These results together suggest that p-value itself is not accurate and it is better to use FDR values to control errors.

MPTGA and TGCT share similarity with common genetic association methods and temporal causality methods. Other methods have been developed to integrate genetic, gene expression and temporal information to construct global regulatory networks[28,42]. Instead of focusing on inferring individual regulations, MPTGA is mainly for genetic association and TGCT is powerful for identifying gene-by-perturbation interactions. Brodt et al.[43] proposed the DyVER method[44], which analyzes genetic effect at each individual time point first, then discretize genetic effects at different time points into two states. Our simulation results indicate that the MPTGA method is more sensitive than methods considering individual time points separately when variances were not independent (Fig. 2). More importantly, when there are missing data, there is no performance degradation for methods modeling all time points together, but there is clear performance degradation for methods

considering individual time points separately (Supplementary Fig. 5). Comparing the results of Brodt et al.[43] and our results, all modules identified in Brodt et al.'s Fig. 5A were also identified by all methods (Table 2), except the ChrIII MATalpha hotspot. Both the static and Fisher methods identified the ChrIII MATalpha hotspot, but the number of traits linked to the locus was less than the cutoff for the MPTGA (Fig. 3 shows that there is a small peak at the locus). On the other hand, multiple teQTL hotspots identified by the MPTGA method are not reported in Brodt et al. For example, the ChrIX70,000 hotspot includes gene expression with genetic effects gradually changing over time (Fig. 4a). Considering the last time point alone, the Wilcoxon rank-sum test $p$-value was < 0.01 (Fig. 4d), but not significantly at the genome level. The DyVER method[44] (as well as the method proposed by Francesconi and Lehner[45]), which aims to group genetic effects at each individual time point into two discrete states is unlikely to work well in the cases with moderate genetic effect changes over time. This also highlights the advantage of the MPTGA method that simultaneously takes all time points into consideration.

In the current study, MPTGA and TGCT are simplified based on a haploid system. They can be generalized for diploid systems (detailed in Methods). When applying the MPTGA and TGCT to diploid systems in which there are three possible genotypes, _00/01/11_ (or _0/1/2_), at each SNP, we can apply these methods directly to detect dominant/recessive effects. To detect full genetic effects, we can estimate parameters for each genotype, then compare them with the estimated parameters without considering genotype (null model). In addition, in the current TGCT test, we explicitly modeled the causal variable in an AR form. If long time-series data are available, a more flexible model can be used to unify the models used in MPTGA and TGCT, such as polynomial functions in both tests (detailed in Methods).

The integration of both genetic and temporal information in our study represents only the beginning step needed to dissect the dynamic regulation. There are also many other directions for improvement in temporal-genetic data analysis. First, other types of available high-throughput data have not been integrated in the analysis yet. To integrate multiple types of data, Zhu et al.[3] reconstructed causal networks and predicted the causal regulators for the eQTL hotspots of gene expression activity in a segregating yeast population. Second, the accuracy of MPTG depends on the amount of data available and data associated measurement errors. The integration of other types of high-throughput data might reduce the influence of these errors. Furthermore, the proposed TGCT method could only address the relationship between a pair of gene expression traits and a locus. More complicated models might be further considered to assess and represent more comprehensive regulation relationships as a larger network, e.g., multiple QTLs affect the expression of multiple transcripts and these RNAs in turn act on another complex trait. Finally, our procedure focuses on identifying causal and reactive relationships which is a very simplistic view of the gene networks. However, the true biology is much more complicated. The genes interacting in a large network may be subject to negative and positive feedback control. Despite these issues, the ability to integrate both the genetic and temporal information in the eQTL analysis offers a promising approach to understand the dynamic regulation.

In practice, the power of MPTGA and TGCT is limited by the number of time points observed, the number of individuals included in a study, and confounding factors. Comparing with the yeast experiments where all experimental conditions are carefully controlled to be similar, there are multiple confounding factors that may contribute to variations to a human time course study, such as age, amount of sleep or physical exercise, food or drug taken, and diseases. Furthermore, genetic architecture of

human is more complex than that of yeast. Blood is the most accessible human tissue for temporal-genetic studies. We previously studied genetic regulation of human blood transcriptome at static state[11] and in time series[24,46]. A large amount of trans-eSNPs for human blood transcriptome were identified using 1002 subjects[11]. Only cis-eSNPs but no trans-eSNP was identified with 40 subjects[46], suggesting it was underpowered for detecting trans-eSNPs. An eQTL study in Japanese population[47] indicates that trans-eSNPs can be detected with 76 subjects. To identify more trans-eSNPs, more subjects are needed in genetic studies. We previously showed that it is possible to infer temporal causal relationships in transcription regulation of human blood transcriptome using 7 time points[24]. To effectively apply the temporal-genetic association and causality tests to a human study, we estimate that at least 8 time points and 200 individuals are needed. It is worth noting that the time intervals in a time series are not needed to be the same. For example, given a polynomial function, we sampled five time points at random intervals and simulated 100 traits (Supplementary Fig. 19a). Then, we fit the simulated traits to a cubic polynomial function, which almost perfectly matches with the pattern underlying the simulated traits (Supplementary Fig. 19b). Thus, when designing a temporal experiment, it is better to sample more time points around the time when derivatives of temporal patterns change.

## Methods

**Yeast data.** A set of time-series messenger RNA gene-expression data is available, which measured the gene expression levels of 95 genotyped haploid yeast F2 segregants after a perturbation with the macrolide drug rapamycin[28]. These segregants were constructed and genotyped by Brem et al.[48] and were derived from two genetically diverse parental yeast strains BY4716 (BY) and RM11-1a (RM). Each yeast segregant in this set of time-series data was sampled at 10 min intervals for up to 50 min after rapamycin addition, and RNA was extracted and profiled with Affymetrix Yeast 2.0 microarrays. This dataset was used for constructing predictive networks by taking advantage of both genetic variations and time dependencies[28,42]. A total 5703 gene expression traits and 2956 SNPs were used in the current study.

**Methods for time series-based eQTL analysis.** To model the behavior of a quantitative trait over time with respect to a given genetic locus, we can model the behavior of the trait using different continuous functions for each possible genotype at a given locus. For example, in the case of a haploid organism, consider a gene expression trait, Y, assayed in individual $i$ at time $t$ at a particular marker location with two genotypes. In this case we can generally represent the expression levels of the trait as $y_i(t) = \delta_{i0}g_0(t) + \delta_{i1}g_1(t) + \varepsilon_i(t)$, where $\delta_{i0}$ and $\delta_{i1}$ are the indicator variables for the two possible genotypes at the marker for individual $i$, and $g_0(t)$ and $g_1(t)$ are functions representing the dynamic process for individuals with different genotypes (the two possible genotypes here have been encoded as _0_ and _1_).

Although the functions could take on any form that can be appropriately parameterized (e.g., exponential, polynomial, and so on), we consider $K$ degree polynomial forms here given they are flexible and commonly used in fitting complex curves: $g(t) = \sum_{k=0}^{K} \beta_k t^k$, with coefficient $\beta_k$ for the exponent $k$. Given this form of trait behavior over time, the trait with respect to a given genetic locus can be expressed as: $y_i(t) = \delta_{i0} \sum_{k=0}^{K} \beta_{k0} t^k + \delta_{i1} \sum_{k=0}^{K} \beta_{k1} t^k + \varepsilon_i(t)$.

For each genetic association approach described below, FDR was estimated by permutation tests in which the strain labels are randomly permuted so that the correlation of the expression traits was maintained, while any genetic associations were destroyed[49].

When applying to the yeast data set, we divided whole yeast genome into 602 bins of 20 kb in size. The thresholds for declaring eQTL hotspots are based on binomial test $p$-value cutoff 0.05/602. We assumed that the number of eQTL in a bin follows the binomial distribution with parameters $n$ = total number of linkage identified across the whole genome and $p$ = 1/602, which assumes equal probability of linkage among the 602 bins. Thus, under the hypothesis of binomial distribution, the threshold $N_0$ was selected such that the probability of observing at least $N_0$ eQTL linkage is less than 0.05/602.

**Static method.** Traditional eQTL analysis is restricted to gene expression levels at a static state, among which a straightforward method is to split segregants into two groups according to their genotypes at a marker and perform the $t$-test or Wilcoxon rank-sum test to check whether there is sufficient evidence that the gene expression levels are significantly different between the two groups.

**Union method**. A straightforward approach to leverage gene expression data of whole time series is to perform eQTL analysis at each time point independently, then combine the results from analysis of all six time point at a locus as the following: $p_j = \text{argmin}_t\, p_{t,j}$, where $p_{t,j}$ is the Wilcoxon rank-sum test $p$-value for the gene expression trait $j$ at the time point $t$. It means that if a gene expression trait was significantly linked to a locus at any of the six time points, the trait was linked to the locus in the union method. The significant $p$-value cutoff is determined by permutation tests with FDR < 0.05.

**Meta-analysis: Fisher's $p$-value**. A meta-analysis approach over a time-series data assumes that data at different points are repeated measurements of the same underlying data. Similar to the union method, we perform eQTL analysis at each time point independently, then combine the results from analysis of all six time point at a locus as the following: $p_j = \prod_t p_{t,j}$, where $p_{t,j}$ is the Wilcoxon rank-sum test $p$-value for the gene expression trait $j$ at the time point $t$. The significant $p$-value cutoff is determined by permutation tests with FDR < 0.05.

**Multivariate analysis of variance**. MANOVA takes into account the covariance between multiple dependent variables and thus is specifically appropriate in testing for association between a SNP and multiple correlated gene expression traits across different time points. In particular, we test the hypothesis:

$$H_0 : \mu_{0t} = \mu_{1t}, \text{ for all } t$$

vs.

$$H_1 : \mu_{0t} \neq \mu_{1t}, \text{ for at least one } t$$

Where $\mu_{gt}$ represents the mean gene expression level in the genotype $g$ group at the testing locus at time point $t$. If MANOVA identifies significant difference of gene expression levels across different time points between groups of samples with different genotypes, we declare an eQTL for the trait at the testing locus.

**Regression method**. Many gene expression changes in time-series were not monotonic and sometimes have more than one fluctuation (Supplementary Fig. 20). Neither linear function nor quadratic polynomial was sufficient to capture underlying these dynamic patterns. On the other hand, there were only 6 time points in our time-series data. A cubic polynomial was sufficient to capture all dynamic patterns in this study (Supplementary Fig. 21). We also tried to use higher degree polynomials to fit the dynamic gene expression changes. The average mean squared errors were similar to the one with the cubic polynomials used (Supplementary Fig. 22). Therefore, we selected the cubic polynomial curve fitting throughout this paper: $g_j(t) = \beta_{0j} + \beta_{1j}t + \beta_{2j}t^2 + \beta_{3j}t^3$. A general regression model with cubic polynomial fitting for a trait $y_i$ is $y_i(t) = \beta_0 + \beta_1 t + \beta_2 t^2 + \beta_3 t^3 + \varepsilon_i$, in which the predictor variables are $(t, t^2, t^3)$. Thus, each set of time-series data contributed six observations in the regression model and the total number of observations was $6N$, where $N$ is the total number of segregants in the yeast F2 data set. To examine the difference between segregants with different genotypes, we compared the reduced model $H_0$ (single fitting $y_i(t) = \beta_0 + \beta_1 t + \beta_2 t^2 + \beta_3 t^3 + \varepsilon_i$) with a full model $H_1$ (separate fitting for each genotype) as

$$y_i(t) = \delta_{i0}(\beta_{00} + \beta_{10}t + \beta_{20}t^2 + \beta_{30}t^3) + \delta_{i1}(\beta_{01} + \beta_{11}t + \beta_{21}t^2 + \beta_{31}t^3) + \varepsilon_i,$$

where $\delta_{i0}$ and $\delta_{i1}$ are the indicator variables for genotype 0 and genotype 1. We performed an F-test to compare the reduced model against the full model to detect eQTL association.

**Multivariate polynomial temporal genetic association**. MPTGA was similar to the regression model described above. Regression model assumed variances at each time points were independent while MPTGA assumes variances are related. Similar to Ma et al.[25], we assumed that for each genotype, the time-series gene expression trait followed a multivariate normal density, in which the mean vector is modeled by a polynomial function $\mathbf{g}_j = \left[g_j(t)\right]_{1 \times m} = \left[\sum_{k=0}^{K} \beta_{kj}t^k\right]_{1 \times m}$, where $m$ is the number of time points in time-series data. Variance at each time point follows a first order AR model AR(1)[50,51] as: $\Sigma = \sigma_e^2 \begin{bmatrix} 1 & \rho & \cdots & \rho^{m-1} \\ \rho & 1 & \cdots & \rho^{m-2} \\ \cdots & \cdots & \cdots & \cdots \\ \rho^{m-1} & \rho^{m-2} & \cdots & 1 \end{bmatrix}$. The density for time-series data could be written as $f_j(\mathbf{y}) = \frac{1}{(2\pi)^{m/2}|\Sigma|^{1/2}}\exp\left[\left(\mathbf{y} - \mathbf{g}_j\right)\Sigma^{-1}\left(\mathbf{y} - \mathbf{g}_j\right)^T/2\right]$.

In our studies, the mean vector was then modeled by the cubic curve $\mathbf{g}_j = [g_j(t)]_{1 \times m} = [\beta_{0j} + \beta_{1j}t + \beta_{2j}t^2 + \beta_{3j}t^3]_{1 \times m}$. The joint likelihood for $N = 95$ segregants was then $L(\Theta) = \prod_{i=1}^{N}\left[\delta_{i0}f_0(\mathbf{y}_i) + \delta_{i1}f_1(\mathbf{y}_i)\right]$, where $\Theta = (\beta_{0j}, \beta_{1j}, \beta_{2j}, \beta_{3j}, \rho, \sigma_e^2)$ is the set of unknown parameters in the statistical model. Maximum likelihood estimates (MLEs) were calculated by taking derivative of $\log L(\Theta)$ with

respect to each unknown parameter. To solve these equations, we first expressed $\beta$'s, $\sigma_e^2$ and $\log L(\Theta)$ as functions of $\rho$ as below, then looked for the critical point of $\log L(\Theta)$ which reached its maximum.

Notation: $\mathbf{T}_0 = \sum_{i=1}^{N}\delta_{i0}\mathbf{y}_i$, $\mathbf{T}_1 = \sum_{i=1}^{N}\delta_{i1}\mathbf{y}_i$, $\mathbf{I}_0 = [1\cdots1]_{1 \times m}$, $\mathbf{I}_1 = [1\cdots m]$, $\mathbf{I}_2 = \mathbf{I}_1^2 = [1\cdots m^2]$, $\mathbf{I}_3 = \mathbf{I}_1^3 = [1\cdots m^3]$, $Q(\rho, \mathbf{U}, V) = \frac{1}{1-\rho^2}\left(U_1V_1 + U_mV_m\right) - \frac{\rho}{1-\rho^2}\left[\sum_{i=1}^{m-1}\left(U_iV_{i+1} + U_{i+1}V_i\right)\right] + \frac{1+\rho^2}{1-\rho^2}\sum_{i=2}^{m-1}U_iV_i$, where $\mathbf{U} = [U_1,\cdots, U_m]$ and $\mathbf{V} = [V_1, \cdots, V_m]$.

By taking derivative of $\log L(\Theta)$ with respect to $\beta_{\cdot 0}$'s, the following linear system was obtained:

$$\begin{bmatrix} \alpha_{11}\beta_{00} + \alpha_{12}\beta_{10} + \alpha_{13}\beta_{20} + \alpha_{14}\beta_{30} = b_1 \\ \alpha_{21}\beta_{00} + \alpha_{22}\beta_{10} + \alpha_{23}\beta_{20} + \alpha_{24}\beta_{30} = b_2 \\ \alpha_{31}\beta_{00} + \alpha_{32}\beta_{10} + \alpha_{33}\beta_{20} + \alpha_{34}\beta_{30} = b_3 \\ \alpha_{41}\beta_{00} + \alpha_{42}\beta_{10} + \alpha_{43}\beta_{20} + \alpha_{44}\beta_{30} = b_4 \end{bmatrix},$$

where $\alpha_{ij} = n_0 Q(\rho, I_{i-1}, I_{j-1})$ and $b_i = Q(\rho, \mathbf{T}_0, I_{i-1})$. Here, $n_0 = \sum_{i=1}^{N}\delta_{i0}$ and $n_1 = \sum_{i=1}^{N}\delta_{i1}$. Then, the coefficients for the linear system could be obtained by

$$\begin{bmatrix} \beta_{00} \\ \beta_{10} \\ \beta_{20} \\ \beta_{30} \end{bmatrix} = \begin{bmatrix} \alpha_{11} & \alpha_{12} & \alpha_{13} & \alpha_{14} \\ \alpha_{21} & \alpha_{22} & \alpha_{23} & \alpha_{24} \\ \alpha_{31} & \alpha_{32} & \alpha_{33} & \alpha_{34} \\ \alpha_{41} & \alpha_{42} & \alpha_{43} & \alpha_{44} \end{bmatrix}^{-1} * \begin{bmatrix} b_1 \\ b_2 \\ b_3 \\ b_4 \end{bmatrix}.$$

$\beta_{\cdot 1}$'s could be obtained similarly. Taking derivative with respect to $\sigma_e$, we had $\sigma_e^2 = \frac{\sum_{i=1}^{N}Q(\rho, \mathbf{y}_i, \mathbf{y}_i) + \sum_{i=0,1}\left[Q(\rho, \mathbf{T}_i, \mathbf{T}_i) - \sum_{j=0}^{3}2\beta_{ji}Q(\rho, \mathbf{T}_i, \mathbf{I}_j) + \sum_{j=0}^{3}\sum_{k=0}^{3}n_i\beta_{ji}\beta_{ki}Q(\rho, \mathbf{I}_j, \mathbf{I}_k)\right]}{mN}$. Since we already had $\beta$'s in terms of $\rho$, here $\sigma_e^2$ was expressed as a function of $\rho$, too. The log likelihood could be written as $\log L(\Theta) = -\frac{mN}{2}\log 2\pi - \frac{N}{2}\left[(m-1)\log(1 - \rho^2) + m\log\sigma_e^2\right] - \frac{mN}{2}$, thus the MLE $\hat{\rho}$ could be obtained by looking for the critical point that maximizes $\log L(\Theta)$. Then the MLE for $(\beta_{0j}, \beta_{1j}, \beta_{2j}, \beta_{3j}, \sigma_e^2)$ could also be obtained.

After determining parameters with MLE procedure, LR test was performed to test the hypothesis of the existence of eQTL by comparing a reduced model $H_0$ (single gene-expression trait curve) against the full model $H_1$ (different gene expression trait curve for different genotypes):

$$H_0 : \beta_{00} = \beta_{01}, \beta_{10} = \beta_{11}, \beta_{20} = \beta_{21}, \beta_{30} = \beta_{31}$$
$$H_1 : \text{at least one of the equalities does not hold.}$$

It is noteworthy that MPTGA is equivalent to the regression method when the AR coefficient $\rho$ is forced to be zero, assuming independent relationship among observations in the time series as the regression method. Therefore, it is expected that the regression method would have similar performance as the MPTGA method when the time-series data is of low self-dependency.

**The AR model**. Time-series data are commonly modeled by a time-lagged AR model (first ordered AR model as an example):

$$Y_{i,t} = \beta_0 + \beta_1 Y_{i,t-1} + \varepsilon_{i,t}.$$

To access whether Trait Y is associated with a genetic locus, we compared a null model

$$H_0 : Y_{i,t} = \beta_0 + \beta_1 Y_{i,t-1} + \varepsilon_{i,t}, t = 2,\cdots, 6$$

vs. a full model (fitting each genotype separately) as

$$H_1 : Y_{i,t} = \delta_{i0}(\beta_{00} + \beta_{10}Y_{i,t-1}) + \delta_{i1}(\beta_{01} + \beta_{11}Y_{i,t-1}) + \varepsilon_{i,t}, t = 2,\cdots, 6,$$

where $\delta_{i0}$ and $\delta_{i1}$ are the indicator variables for genotype 0 and genotype 1. It is noteworthy that this formulation corresponds to the independent model M3 in the TGCT test section below and we will refer to this method as the AR method in temporal-genetic association tests. We performed a linear regression to estimate the parameters under each model and used an F-test to compare the null model against the full model to detect eQTL associations.

**Estimating confident intervals of eQTLs**. We employed the $\chi^2$ quantile method in the LOD score test described in Mangin et al.[52], in which the corresponding statistic $T(d_0)$ follows a chi square distribution with $N$ degree of freedom under the null hypothesis that $d_0$ is the QTL position. The $(1 - \alpha)$ confidence interval is then defined as $[d_{\text{inf}}, d_{\text{sup}}]$, where $d_{\text{inf}}$ ($d_{\text{sup}}$) is the smallest (the greatest) value of $d_0$ such that $T(d_0)$ is smaller than $\chi^2_{N,\alpha}$, where $\chi^2_{N,\alpha}$ is the $\alpha$ quantile of a $\chi^2$ with $N$ degree of freedom. Here, $T(d_0) = \sup_d[R(d)] - R(d_0)$ and $R(d)$ is the $-2*$log-LR statistic $R(d) = -2\log\frac{\text{likelihood of data with no eQTL}}{\text{likelihood of data with an eQTL at } d}$.

**Temporal-genetic causality test**. Temporal QTL can be treated as a systematic source of perturbation to infer causality among traits associated with the QTL. We and others have previously demonstrated that for two traits associated with a given genetic locus there are a limited number of causal relationships possible between the traits[14,17] (Supplementary Fig. 1): (1) Trait $X$ is causal for Trait $Y$ (M1); (2) Trait $Y$ is causal for Trait $X$ (M2); (3) Trait $X$ is independent of Trait $Y$ (M3); (4) Trait $X$ is partially causal for Trait $Y$ (M4); (5) Trait $Y$ is partially causal for Trait $X$ (M5). Models M1 and M2 are the simplest causal relationships between two traits in which a given locus acts on one of the traits through the other. Model M3 is the fully independent model in which the genetic locus acts independently on each trait. Models M4 and M5 represent partial causal relationships in which one trait is causal for the other, but the genetic locus acts independently on each trait.

Static eQTLs and teQTLs were not evenly distributed along the whole genome. There were loci referred to as eQTL hotspots where many gene expression traits were linked. It is important to dissect causal regulators underlying these eQTL hotspot loci, which can regulate a large number of gene expression traits. To identify causal regulators for a given hotspot, Zhu et al.[3] first identified genes with cis-eQTL in the corresponding eQTL hotspot region and inferred their downstream-regulated genes as the set of genes that could be reached in the integrative molecular Bayesian Network. If the downstream set of a cis regulated gene at an eQTL hotspot locus is significantly enriched for eQTLs linked to the locus, the cis regulated gene is inferred as a key regulator of the eQTL hotspot. Instead of integrating diverse data into a global causal network[3,28,42,53], we aim to test pairwise causality by leveraging time-dependent genetic data.

The LCMS proposed by Schadt et al.[14] used normal distributions to model the static time expression trait data. Here with multi-dimensional time-series data, we seek to combine both the dynamic information and genetic information to infer the causal relationship between two time series more precisely. Granger[26] formalized the idea of time series-based causality test in the context of linear regression. The idea of Granger causality is to test whether the prediction of the time series could be significantly improved by incorporating information from previous time points in a second time series, and thus to test whether the second time series has a causal effect on the first time series. Mathematically, Granger causality test compares the reduced model with the full model, which adds the lagged information of another time series as a predictor in regression, and tests whether the improvement in fitting the data is significant. We adopted the idea to include the lagged values of one time series to augment the autoregression when comparing the causal relation and independent relation. Due to a small number of time points available, we used first-order autoregression model AR(1). Specifically, the five models in Supplementary Fig. 1 were represented as:

$$M1 : X_{i,t} = \delta_{i0}(\alpha_{00} + \alpha_{10}X_{i,t-1}) + \delta_{i1}(\alpha_{01} + \alpha_{11}X_{i,t-1}) + \varepsilon_{i,t}$$
$$Y_{i,t} = \beta_0 + \beta_1 Y_{i,t-1} + \beta_2 X_{i,t-1} + \mu_{i,t}$$
$$M2 : Y_{i,t} = \delta_{i0}(\beta_{00} + \beta_{10}Y_{i,t-1}) + \delta_{i1}(\beta_{01} + \beta_{11}Y_{i,t-1}) + \mu_{i,t}$$
$$X_{i,t} = \alpha_0 + \alpha_1 X_{i,t-1} + \alpha_2 Y_{i,t-1} + \varepsilon_{i,t}$$
$$M3 : X_{i,t} = \delta_{i0}(\alpha_{00} + \alpha_{10}X_{i,t-1}) + \delta_{i1}(\alpha_{01} + \alpha_{11}X_{i,t-1}) + \varepsilon_{i,t}$$
$$Y_{i,t} = \delta_{i0}(\beta_{00} + \beta_{10}Y_{i,t-1}) + \delta_{i1}(\beta_{01} + \beta_{11}Y_{i,t-1}) + \mu_{i,t}$$
$$M4 : X_{i,t} = \delta_{i0}(\alpha_{00} + \alpha_{10}X_{i,t-1}) + \delta_{i1}(\alpha_{01} + \alpha_{11}X_{i,t-1}) + \varepsilon_{i,t}$$
$$Y_{i,t} = \delta_{i0}(\beta_{00} + \beta_{10}Y_{i,t-1}) + \delta_{i1}(\beta_{01} + \beta_{11}Y_{i,t-1}) + \beta_2 X_{i,t-1} + \mu_{i,t}$$
$$M5 : Y_{i,t} = \delta_{i0}(\beta_{00} + \beta_{10}Y_{i,t-1}) + \delta_{i1}(\beta_{01} + \beta_{11}Y_{i,t-1}) + \mu_{i,t}$$
$$X_{i,t} = \delta_{i0}(\alpha_{00} + \alpha_{10}X_{i,t-1}) + \delta_{i1}(\alpha_{01} + \alpha_{11}X_{i,t-1}) + \alpha_2 Y_{i,t-1} + \varepsilon_{i,t}$$

We used different autoregression parameters for different genotypes to account for the genetic effect and added the lagged value of one time series to represent the causal effect of one time series on the other. The parameters in each model were estimated using ordinary linear regression. The log likelihood of $X$ and $Y$ are calculated as

$$\ln(\hat{L}(\mathbf{X})) = -\frac{N}{2}\ln(2\pi) - \frac{N}{2}\ln\left(\sum_{i,t>1}\left(X_{i,t} - \hat{X}_{i,t}\right)^2\right) + \frac{N}{2}\ln(N) - \frac{N}{2}$$

and

$$\ln(\hat{L}(\mathbf{Y})) = -\frac{N}{2}\ln(2\pi) - \frac{N}{2}\ln\left(\sum_{i,t>1}\left(Y_{i,t} - \hat{Y}_{i,t}\right)^2\right) + \frac{N}{2}\ln(N) - \frac{N}{2}$$

When assessing the causal (M1: $X \rightarrow Y$) and reactive (M2: $Y \rightarrow X$) models, we calculated log joint likelihood $\ln\hat{L}(\mathbf{X}, \mathbf{Y}) = \ln\hat{L}(\mathbf{X}) + \ln\hat{L}(\mathbf{Y})$ under these two models. As the total numbers of parameters in M1 and M2 are the same, comparing $\ln\hat{L}(\mathbf{X}, \mathbf{Y})$ under these two models and comparing BICs are equivalent.

One of our major goals of the TGCT test is to identify the cis-regulators of teQTL hotspots. If we assume Trait $X$ with a cis-eQTL linked to a teQTL hotspot, then we can restrict the model selection among causal (M1: $X \rightarrow Y$), independent (M3: $X \perp Y$), and partial causal (M4) models without considering the reactive (M2: $Y \rightarrow X$) and partial reactive (M5) models. In such cases, the three models share the same regression model for Trait $X$. Thus, we perform model selection based only on the regression on Trait $Y$. The corresponding log likelihood was estimated as

follows:

$$\ln(\hat{L}) = -\frac{N}{2}\ln(2\pi) - \frac{N}{2}\ln\left(\sum_{i,t>1}\left(Y_{i,t} - \hat{Y}_{i,t}\right)^2\right) + \frac{N}{2}\ln(N) - \frac{N}{2}$$

And BIC is defined as $\text{BIC} = \ln(N)k - 2\ln(\hat{L})$, where $k$ is the number of parameters estimated in the corresponding model. BIC penalizes complex models. The model with the smallest BIC was identified as the model best supported by the data.

For each teQTL hotspot, we first identified genes with cis-teQTLs linked to the hotspot as candidate causal genes, then pair these cis-eQTL genes with all genes with trans-eQTLs linked to the hotspot for the causality test. The cis-eQTL genes with the number of causal relations significantly more than expected by chance (the cutoff value for defining a teQTL hotspot) were selected as the putative key regulators of the eQTL hotspot.

**MPTGA and TGCT in diploid systems**. The above MPTGA and TGCT are simplified based on a haploid system. When applying the MPTGA and TGCT to diploid systems in which there are three possible genotypes, 00/01/11 (or 0/1/2), at each SNP, we can apply these methods directly to detect dominant/recessive effects. To detect full genetic effects, the genetic association test can be expressed as $y_i(t) = \delta_{i0}\sum_{k=0}^{K}\beta_{k0}t^k + \delta_{i1}\sum_{k=0}^{K}\beta_{k1}t^k + \delta_{i2}\sum_{k=0}^{K}\beta_{k2}t^k + \varepsilon_i(t)$ for a given trait $Y$, then the reduced model $H_0$ (single-gene expression trait curve):

$$\beta_{k0} = \beta_{k1} = \beta_{k2} \text{ for all } k$$

can be compared against the full model $H_1$ (different gene expression trait curve for different genotypes): at least one of the equalities does not hold.

To test the hypothesis of the existence of eQTL at a locus, we can estimate these parameters with an MLE procedure and performing LR test as we described in the above section. Similar generalization can be applied to the TGCT.

**A generalized TGCT**. One potential drawback in the current TGCT test is that we explicitly modeled the causal variable in an AR form, which is not as powerful in identifying genetic effects as other methods (Supplementary Fig. 4). This also leads to the results that the proposed temporal-genetic association test (MPTGA) and the causality test (TGCT) are based on two different forms of models instead of a unified function. If long time-series data are available, a more flexible model can be used to unify the approaches used in temporal-genetic association and causality tests. Specifically, the models can be specified as follows: given Trait $X$ with cis-eQTL, $X_{it} = \delta_{i0}f_{x0}^{t-1}(t) + \delta_{i1}f_{x1}^{t-1}(t) + \varepsilon_t$, and Trait $Y$ with trans-eQTL, then the three possible models of the causal relationships between them can be rewritten as the following

$$M1(\text{causal model}) : Y_{i,t} = f^{t-1}(t) + \beta_2 X_{i,t-1} + \varepsilon_t$$
$$M3(\text{independent model}) : Y_{i,t} = \delta_{i0}f_0^{t-1}(t) + \delta_{i1}f_1^{t-1}(t) + \varepsilon_t$$
$$M4(\text{partial causal model}) : Y_{i,t} = \delta_{i0}f_0^{t-1}(t) + \delta_{i1}f_1^{t-1}(t) + \beta_2 X_{i,t-1} + \varepsilon_t,$$

Where $f_0^{t-1}(t)$ and $f_1^{t-1}(t)$ correspond to polynomial fitting functions using previous time points of each genotype, respectively; $f^{t-1}(t)$ corresponds to a single polynomial fitting function using previous time points of both genotypes. Then, we can test for temporal-genetic causality using the same model selection approach as described in the TGCT method. We can set the degrees of polynomial functions $f^{t-1}(t)$, $f_0^{t-1}(t)$, and $f_1^{t-1}(t)$ as the same as the polynomial function $f(t)$ used in MPTGA. If the number of time points is not large enough for the unified model described above, but larger than the size in our current study, we can make TGCT more flexible by using higher order AR models instead of first-order AR models.

**Permutation tests**. Two types of information are critical to temporal-genetic association tests: (1) temporal relationships; (2) genetic structures (LD structures across the genome). Thus, in the permutation procedure, we preserved the temporal relationships and genetic structure, and only permuted the strain labels. We left the gene expression data unchanged and permuted the strain labels in the genetic data so that true generic associations were destroyed while the correlation relationship of expression traits was maintained. We performed the permutation 10 times. At a specific $p$-value cutoff, the FDR was calculated as: $\text{FDR} = \frac{\text{average \# associations} < p \text{ in permuted data}}{\text{\# associations} < p \text{ in original data}}$. The $p$-values cutoffs needed to control the FDR at the 5% level were used in our temporal-genetic association tests.

**Simulation studies for temporal genetic associations**. We simulated time-series data sets from multivariate normal distribution, with mean vector modeled by various patterns that are similar to the observed experimental results (Supplementary Fig. 2). Each set of data was then drawn either from a single model or two separate models with equal probability, which mimic the situation of existence and absence of eQTL effects. Ten thousand data sets were simulated, in which each data set consisted of $N$ six-point time series either from a single multivariate normal

distribution or two separate multivariate normal distributions. The number of samples $N$ varied from 20 to 100. The covariance matrix was modeled as above, where $\rho$ was between 0.1 and 0.9 with $\rho \sim N(0.9, 0.02)$ for high-correlation data set or $\rho \sim N(0.1, 0.02)$ for low-correlation data set.

**Robustness of temporal-genetic association methods**. We assessed the robustness of temporal-genetic association methods by randomly dropping data points in the simulated time series with data missing rate varying from 0.02 to 0.1. For methods that involve fitting a curve to the data within each genotype, i.e., MPTGA, regression and AR, the samples with missing time points were masked first, then each method was applied to the remaining samples (corresponding forms of curves fitted to the remaining data), then the missing time points were imputed based on the fitted curves and the temporal-genetic association methods were applied to the imputed data. For the other methods that do not fit curves to the data, i.e., union, Fisher and MANOVA, the samples with missing data were masked first and each method was applied to the remaining data.

**Simulation studies for TGCT**. To evaluate the performance of the TGCT test, we simulated pairs of time series for traits $X$ and $Y$ under different models. We performed two sets of simulation studies. In the first set of studies, we simulated 10,000 trait pairs for each parameter setting. Each Trait $X$ consisted of 6 time points for $N$ samples with the mean vectors following one of the patterns shown in Supplementary Fig. 2. The genetic effects were simulated by drawing variations from two separate multivariate normal distributions. Each Trait $Y$ was simulated from Trait $X$ according to the causal model (M1). The covariance matrices were modeled similarly with $\rho \sim N(0.8, 0.1)$ for each dataset. The above simulation scheme was repeated to generate 10,000 trait pairs for each sample size $N$ that varied from 20 to 150 and with each set of parameters in the causal model M1. For the pairs with both traits linked to the tested locus (MPTGA $p$-value $< 10^{-6}$), we compared the joint likelihood $L(X,Y)$ based on the causal model (M1) with the reactive model (M2). In the second set of studies, we simulated 10,000 trait pairs. Each Trait $X$ consisting of six time points for $N$ samples was simulated similar as above, and Trait $Y$ was simulated according to the causal (M1), independent (M3), or partial causal (M4) model with different parameter settings. For the pairs with both traits linked to the tested locus (MPTGA $p$-value $< 10^{-6}$), we calculated the likelihoods of Y based on the causal (M1), independent (M3), and partial causal (M4) models, and selected the best fit model based on BIC (detailed in the Methods section above).

**Assessing overfitting problem**. Both the MPTGA and the regression method, or the polynomial regression based methods in general, are prone to sporadic associations or overfitting[37]. To assess the tendency of sporadic association or overfitting in the MPTGA or the regression method, we compared $p$-values of significant associations in both empirical and permuted data and the $p$-values of neighboring SNPs that are in strong LD. If a significant trait-SNP association is detected (a statistical model is trained) and the trained model describes the true underlying relationship, then neighboring SNPs in high LD, where genotypes for SNPs in high LD data vary slightly (according to the strength of the LD structure), should be able to predict the trait or strongly associate with the trait (model testing). On the other hand, if a significant trait-SNP association detected (a statistical model is trained) and the trained model describes noise instead of the true underlying relationship, then neighboring SNPs in high LD are unlikely to be able to predict the trait or strongly associate with the trait (model testing). Thus, by comparing the consistency/correlation of the strengths of associations between peak SNPs and neighboring SNPs in high LD to a trait we can assess overfitting problem, less overfitting will lead to a higher consistency/correlation.

**Assessing statistical validity**. To assess the statistical validity of the MPTGA test, we compare the QQ plots from the $p$-values of the MPTGA test based on (1) simulated data, (2) real data, and (3) permuted real data.

The simulation scheme is as follows: first, we simulate a genotype vector for 95 samples with each cell taking a random *0/1* value with 0.5 probability; then we simulate a random gene expression trait (95 samples × 6 time points) from a multivariate normal distribution with a mean vector corresponding to a random pattern in Supplementary Fig. 2 as we use in the Simulation studies for temporal genetic associations section; finally, the MPTGA test is applied to the simulated genotype and gene expression traits. The gene expression matrix is simulated independently from the genotype matrix, this simulation is depicting a scenario with no association between the gene expression trajectories and the genotypes. For the yeast dataset, we tested 5703 gene expression traits v.s. 2956 SNPs, resulting in a $5703 \times 2956$ $p$-value matrix. Next, we permuted the strain labels and performed MPTGA test, resulting in another $5,703 \times 2956$ $p$-value matrix in each permutation.

**RRD1 KO experiments**. The wild type strain BY4730 and RRD1 knockout strain YSC6273-201925697 were obtained from Thermo Scientific Open Biosystems. Yeast was grown in YPD medium to log-phase in shaken flasks at 30 °C. Total RNA was extracted as described previously[54]. For rapamycin treatment, 100 nM rapamycin (Cayman Chemical, Ann Arbor, MI) was added to the medium after yeast grew to log-phase. After culture for 50 min, total RNA was extracted the same as above. All experiments were repeated 3 times on three different days.

Approximately 250 ng of total RNA per sample was used for library construction by the TruSeq RNA Sample Prep Kit (Illumina) and sequenced using the Illumina HiSeq 2500 instrument with 100nt single-read setting according to the manufacturer's instructions. Sequence reads were aligned to yeast genome assembly using Tophat[55]. Total 6932 yeast transcripts were quantified using Cufflinks[55], and 5542 of them overlap with transcripts on Yeast Genome 2.0 Arrays from Affymetrix, which was used for generating the yeast F2 time course data. The 5542 transcripts were used in further analysis. DEGs were defined by CuffDiff[55]. At $q$-value $< 0.01$, 64 and 581 were in *RRD1* ko signature without rapamycin (*RRD1* ko no treatment vs. wild-type no treatment) and *RRD1* ko signature with rapamycin (*RRD1* ko with rapamycin vs. wild type with rapamycin), respectively. The RNA sequencing data generated are available at GEO data base with accession number GSE86786.

**eQTL hotspots**. The yeast genome is divided into 20 kb bins and the number of (t) eQTLs associated with markers in each bin is counted. For those bins with significantly more (t)eQTLs than expected by chance, the genetic location corresponding to the bin is defined as a (t)eQTL hotspot[29,56]. If neighboring bins were (t)eQTL hotspots, then they are merged into a single (t)eQTL hotspot.

**Gene-set enrichment**. The yeast GO categories were derived from the SGD database (http://db.yeastgenome.org/cgi-bin/GO/goTermFinder). We restricted attention to GO terms based on the slim mapping from SGD, which is comprised of roughly 100 categories. We applied the hypergeometric test using the annotation database. The annotations with the most significant $p$-values were reported in Table 3. We also applied the hypergeometric test for enrichment analysis for all signatures comparison.

**Code availability**. Codes for MPTGA and TGCT can be found at http://research.mssm.edu/integrative-network-biology/Software.html.

## Data availability

The RNAseq data generated in this study is available at GEO database with accession number GSE86786.

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

## Acknowledgements

The work is partially supported by R01AG046170, U01HG008451, and U19AI118610.

## Author contributions

L.L. and J.Z. conceived the study and developed models. E.E.S. provided valuable suggestions in model development. J.P.H. performed validation experiments. Q.C. contributed significantly to the simulation studies. K.Y., R.E.B., E.E.S., and J.Z. participated in data collection. L.L., Q.C., S.Y., Z.T., E.E.S., and J.Z. participated in data analysis. L.L., Q.C., E.E.S., and J.Z. contributed in manuscript writing. All authors approved the final manuscript.

## Additional information

**Competing interests:** The authors declare no competing interests.

