## [Peer Review file · Nature Communications]

Reviewers' comments:

Reviewer #1 (Remarks to the Author):

Summary

This work proposes new methods to identify temporal QTLs from time-series data, as well as a subsequent pairwise test for causality. The te-QTL approach extends previous work by modelling autocorrelation between time points through a multivariate model, and the causal inference extends Granger causality testing to include genetic QTL effects. The methods are applied to gene expression from a yeast cross with rapamycin response and show the proposed approach to identify the largest number of QTLs. The causal inference test replicates previous findings and identifies two hot spots with compelling associations, one of which is followed up in a knockout screen.

Overall, I the work is of high quality and a laudable integration of new methods with innovative biological data. The MPTGA test is a logical extension of previous approaches and shown to increase power both in simulation and real data. The identified hotposts and causal validation is compelling and convincingly demonstrates how the methods provide actionable insights. The work offers advances in two key areas - time series modelling and causal inference - that are of great relevance beyond model organism genetics. My main concerns are about the causal inference test, which is not well presented nor justified by simulation. In my opinion its an approach of potentially large interest but reads like an afterthought. I urge the authors to more carefully describe and analyse this test.

Major Comments

1. It is very difficult to understand what the temporal+genetic causality test (TGCT) is doing from the main text. The authors provide sufficient context for the test but the actual statistical model is described in two vague sentences (lines 144-147). I encourage the authors to include some of the mathematical framework in the main text, but at the very least please provide (a) a clear definition of the specific models being evaluated; (b) an explanation of how the underlying parameters are estimated; (c) an explanation of how the tests are applied to hotspots (is every pair of genes in the hotspot evaluated for causality?), how multiple-testing is accounted for, and how the top causal regulator is selected.

2. I do not see any simulation results presented for the TGCT, nor any clear intuition on how the test is expected to perform relative to previous methods or when the underlying models are misspecified. Additional simulations are needed to make these comparisons and show that the method actually has power at the current sample size and data complexity. As the authors point out, previous work in causal inference has often overstated the statistical power to make robust statements about causality so this is a special concern. Some discussion of how TGCT is expected to perform when the underlying causal model is more complex than those considered would also improve the interpretation.

3. The applicability of this work to other studies could be significantly increased by demonstrating some general trends in statistical power. In particular, how well do the proposed methods perform if some time points are randomly dropped from the analysis? How many of the te-QTLs are still identified? And the same for dropping individuals from the analysis. The discussion claims that 8 time points and 200 individuals are needed for human data; these numbers are very important but it's not at all clear how they have been derived.

Reviewer #2 (Remarks to the Author):

The paper is focused on the analysis of genetic associations and genetic causality analysis using temporal gene expression data. Specifically, it suggests a methodology for temporal genetic association called MPTGA and a methodology for temporal genetic causality called TGCT. The methods are demonstrated using synthetic data and data of rapamycin gene expression response across a genotyped yeast F2 population.

Major comments

1. Temporal association analysis: novelty and comparisons. The proposed association method is closely related to Ma et al. In fact the paper use the same likelihood ration test and the only difference is in the way the MLE for the parameters is calculated. In the introduction the authors claim to develop a novel methodology for temporal association, but in light of the close relation to Ma et al, the novelty of the method should be clearer stated (also in introduction) and a direct comparison with Ma et al using synthetic data is required to demonstrate superiority.
2. Temporal Causality analysis: novelty and comparisons. The temporal causality model appears to be the same as the approach of several publications about mediation analysis (e.g., [2,3,4,5]). It seems that the only novelty if the practical translation to genetics. If there are no substantial differences, this should be emphasized. If there is a substantial difference, a comparison to the existing methods in Refs [2-5] should be included. Differences from this literature should be also discussed in the introduction section.
3. The code for MPTGA and TGCT should be released on a repository such as cRAN.
4. Statistical validity of the association with MPTGA. It is important so demonstrate that the p-values derived from the chi-sq approximation (lines 537-543) are calibrated. QQ-plots and inflation parameter are required in both real and synthetic data. The large number of associations in the case of ribosomal proteins suggest spurious associations of responding genes and therefore it is especially important to show that the ribosomal modules are not only due to statistical inflation (see also comments 5,6,8).
5. Statistical validity of the causality analysis with TGCT. The different models are compared using BIC criteria and the best model according to the BIC score is preferred. Several recent papers show that it is preferable to perform genetic causality using more advanced tests (rather than the BIC or AIC criteria), such as Ref [7] (Neto et al). The distribution of likelihood ratios between M1 versus the max of the alternative models (as in [7]) should be also evaluated empirically using reshuffling of gene expression, especially for DDR1 and for the ribosomal genes, since the theoretical assumptions do not necessarily hold in the real scenario.
6. Prior work (see for example Ref [8]) indicate that association hotspots are mainly due to spurious associations. This may be the case here in light of the large number of associations of ribosomal genes. eQTL maps should be demonstrated (as in [8]) and control for spurious associations should be applied if necessary.
7. The relevance to human and mouse data is not clear. Both methods rely on binary SNP of 0/1 values. Heterozygosity is not handled, and extension to inbred mice populations (such as CC and DO) that rely on multiple parental strains is not clear. Relevance and possible extensions should be discussed.
8. Recent prior work [1](Brodt et al.) has already applied temporal eQTL analysis on the same yeast

dataset. It seems that most of the identified eQTLs here have been already published in the previous work. A clear comparison should be included (which modules are the same? which ones are novel and which ones are missing?). From a brief comparison, I noticed that the main contribution here is the identification of associations in ribosomal genes. However, this result may be an artifact, as none of the ribosomal genes were previously identified; this may occur due to inflation of scores in the highly responding ribosomal genes. The validity of ribosomal genes result needs to be addressed via (i) clear visualization of the different dynamics in ribosomal genes with different genetic backgrounds (e.g., as in Figure 4B in Ref [1]), (ii) showing that it is not due to inflation or confounding effects (as mentioned in Q. 4 above), (iii) discuss why the ribosomal associations were not previously identified. As it stands, the interpretation of the ribosomal genes associations—a major part of the actual novelty—is a major concern.

Minor comments:

9. Two recently published temporal association methods are Refs [1] and [6]. It is advantage to compare to these methods.

10. How the methodology handles intraindividual change over time. Is it possible to discriminate between time-specific causal effects from sustained causal effects?

11. The model assumes a 2-time points lag. In the literature a longer lag is also considered.

References

- [1] Dissecting Dynamic Genetic Variation That Controls Temporal Gene Response in Yeast. A Brodt, M Botzman, E David, I Gat-Viks. PLOS Comp Biol 2014.
- [2] Scott E. Maxwell , David A. Cole & Melissa A. Mitchell (2011) Bias in Cross-Sectional Analyses of Longitudinal Mediation: Partial and Complete Mediation Under an Autoregressive Model, Multivariate Behavioral Research, 46:5, 816-841, DOI:10.1080/00273171.2011.606716
- [3] James P. Selig and Kristopher J. Preacher (2009). Mediation Models for Longitudinal Data in Developmental Research. Research in Human Development, 6(2–3), 144–164, 2009
- [4] Magnusson Hanson LL, Peristera P, Chungkham HS, Westerlund H (2016) Longitudinal Mediation Modeling of Unhealthy Behaviors as Mediators between Workplace Demands/Support and Depressive Symptoms. PLoS ONE 11(12): e0169276. doi:10.1371/journal.pone.0169276
- [5] AN EXAMINATION OF STATISTICAL METHODS FOR LONGITUDINAL MEDIATION MODELING, dissertation by BIRMINGHAM, ALABAMA, 2011.
- [6] Francesconi M, Lehner B (2014) The effects of genetic variation on gene expression dynamics during development. Nature 505: 208–211.
- [7] Modeling causality for pairs of phenotypes in system genetics EC Neto, AT Broman, MP Keller, AD Attie, B Zhang, J Zhu, BS Yandell Genetics, 193: 1003-1013, 2013. DOI: <http://dx.doi.org/10.1534/genetics.112.147124>
- [8] Effectively identifying regulatory hotspots while capturing expression heterogeneity in gene expression studies. Jong Wha J Joo, Jae Hoon Sul, Buhm Han, Chun Ye and Eleazar Eskin Genome Biology 2014. DOI: 10.1186/gb-2014-15-4-r61

Reviewer #3 (Remarks to the Author):

The manuscript by Zhu and colleagues studies the issue of mapping eQTL in time-series expression data (temporal eQTL or teQTL), and proposes to combine the benefits of genetic perturbations and

time series information to reconstruct causal relationship between genes. Most eQTL studies were performed on single conditions, but as the authors argued, there is clearly a need of exploring dynamic transcriptome response. Thus, the teQTL methodology described in this work is very interesting, and the results seem to demonstrate the advantage of the method they developed (MPTGA). The authors also studied the problem of identifying causal regulators of gene expression traits, using the idea of Granger causality. However, I have some serious doubt of this part of their work.

Major comments

The authors evaluated their teQTL method, MPTGA, against simpler alternative such as union method and Fisher's meta-analysis. But mapping teQTL can be viewed as a problem of testing association of a SNP with multiple phenotypes. One can simply view expression level at each time point as a trait, and test all traits (i.e. time points) jointly. This problem of cross-phenotype association has been extensively studied before, e.g. see PMID: 23752797 for a review. Thus it would be appropriate to evaluate the performance of these related methods in the teQTL setting. In particular, a common multivariate association method is based on MANOVA (or its variants). Basically, suppose there are K traits, one has $Y_k = \beta_k G + \epsilon$, where G is genotype of the test SNP, and Y_k the k -th trait. The problem is to test $H_0: \beta_1 = \dots = \beta_K = 0$.

The authors compared MPTGA with three other methods in yeast data and showed a higher performance of MPTGA (Table 1). One question is that what is the extent of the overlap among these methods. It is possible that MPTGA finds most of the teQTL identified by other methods, but gain some extra ones. But it is also possible that MPTGA captures somewhat different aspects of data than other method, and thus the overlap of teQTL from different studies would be relatively small. My guess is that MPTGA is good at capturing the overall "shape" of the time series data, but others may be more sensitive to change of expression at only one time point.

The TGCT model (equations from line 581 to 589) looks somewhat strange to me. The authors tries to expand the Granger causality test to include genetic effects. But the way they model genetic effects is completely different from what they have done in the teQTL part. In the TGCT model, a SNP could change the coefficients in the Granger causality model, e.g. α_{00} vs. α_{01} and α_{10} vs. α_{11} in M1. But these coefficients represent the extent of auto-correlation between adjacent time points of an expression trait, while eQTL (or teQTL) measures the genetic effect on actual expression level. For example, the alpha parameters here (say M1) do not change with time, while the eQTL effects are generally time-dependent (say a SNP may affect expression only in one time point). It is difficult to reconcile these very different views of eQTL, and I do not see any evidence at all in this paper, that eQTL would behave in this way (i.e. changing auto-correlation coefficients). This may be related to a comment below about the Table 4 results.

The procedure for finding causal regulator of teQTL hotspots is not described clearly. I would suggest the authors to create a separate subsection in Methods to explain the procedure. Given a teQTL hotspot and its associated genes, one key question I have is: do they search all genes within the hotspot; or only genes that are cis-associated with the hotspot (from teQTL analysis). Another question is: they need to compare the number of causal relations between a cis-gene and the hotspot-associated genes with random expectation, how is this random expectation calculated? Because the hotspot-associated genes have correlated expression patterns, some extra care is needed to obtain the random distribution.

In Table 4, the authors reported a number of candidate causal cis-regulators of teQTL hotspots. The results look suspicious to me. The teQTL hotspots are not large, based on the authors description

(perhaps 10 genes on average, as they have 600 bins). And I would think that most hotspots have 1-2 true causal regulators that mediate the effect of hotspots on the associated genes. But the authors reported much larger number of regulators. The main justification in the manuscript is some experimental and literature evidence of two genes ISC1 and RRD1. We would need much stronger evidence that most of the predicted regulators are real. In light of possible statistical problems (two previous comments), I am concerned that many of them could be false positives.

Minor comments

Are the authors planning to make their software MPTGA widely available? It seems useful. My question is about generality of the software. Can it be used for data with an arbitrary number of times point? In the method presented, the authors fit a cubic polynomial, but to use it in another dataset, a higher degree may be needed. Is there a routine to automatically choose such parameters?

RRD1 has two eQTL, one cis in chrIV: 70K and another trans in chrIV: 150K. There are teQTL hotspots in both loci. The authors said that both hotspots could act be changing RRD1 level, if this is true, the genes associated with both hotspots should have significant overlap. Is this true?

Line 249, 253: Should "Punitive" be "putative"?

We thank the reviewer's constructive suggestions. We addressed reviewers' comments pertaining to our manuscript "*Temporal Genetic Association and Temporal Genetic Causality Methods for Dissecting Complex Networks*", and revised the manuscript accordingly as detailed below. Specifically, we performed multiple simulation studies to evaluate robustness of our temporal-genetic association model and performance of our temporal-genetic causality model. The reviewers' comments are in black font type and our responses are given in blue font type. All page/line numbers and others such as references given are with respect to the revised manuscript unless otherwise stated.

Reviewer #1 (Remarks to the Author):

Summary

This work proposes new methods to identify temporal QTLs from time-series data, as well as a subsequent pairwise test for causality. The te-QTL approach extends previous work by modelling autocorrelation between time points through a multivariate model, and the causal inference extends Granger causality testing to include genetic QTL effects. The methods are applied to gene expression from a yeast cross with rapamycin response and show the proposed approach to identify the largest number of QTLs. The causal inference test replicates previous findings and identifies two hot spots with compelling associations, one of which is followed up in a knockout screen.

Overall, I the work is of high quality and a laudable integration of new methods with innovative biological data. The MPTGA test is a logical extension of previous approaches and shown to increase power both in simulation and real data. The identified hotposts and causal validation is compelling and convincingly demonstrates how the methods provide actionable insights. The work offers advances in two key areas - time series modelling and causal inference - that are of great relevance beyond model organism genetics. My main concerns are about the causal inference test, which is not well presented nor justified by simulation. In my opinion its an approach of potentially large interest but reads like an afterthought. I urge the authors to more carefully describe and analyse this test.

We thank the reviewer for encouraging comments. We performed multiple simulation studies and updated text about the temporal-genetic causality test as the reviewer suggested. Details are listed below.

Major Comments

1. It is very difficult to understand what the temporal+genetic causality test (TGCT) is doing from the main text. The authors provide sufficient context for the test but the actual statistical model is described in two vague sentences (lines 144-147).

We added some details of the temporal-genetic causality test in the main text (Lines 160-178), explaining the general framework of the test, the intuition of the model setting, and how it is applied in the real data analysis as the following:

"More specifically, after identifying two traits X and Y with temporal-genetic association to the same locus, there are five possible causal/reactive relationships as shown in Supplementary

Figure 1. In a causal model (M1: $X \rightarrow Y$), the genetic effect (or the association with the marker) of Trait Y is solely explained by Trait X, so that the time series values of Trait Y can be predicted with values of Traits X and Y at previous time points. In an independent model (M3: $X \perp Y$), the genetic effect of Trait Y cannot be explained by Trait X. In a partial causal model (M4), the genetic effect of Trait Y can only partially be explained by Trait X, so that the time series values of Trait Y can be predicted with values of Traits X and Y at previous time points as well as the genotype information at the associated locus. When traits X and Y were switched in models M1 and M4, the causal and partial causal relationships can be represented in models M2 and M5, respectively. First, we assessed the power to distinguish causal/reactive relationships (M1 vs. M2) in general. We compared joint likelihood $L(X, Y)$ and selected the model better fitting the data (detailed in Methods). Then, we leveraged genetic information to reduce the number of pairs of traits to test, focusing on the cis-trans trait pairs as the following: Trait X has a cis-teQTL and Trait Y has a trans-teQTL at the same locus so that models to be assessed are limited to M1, M3, and M4. We applied a linear regression on the corresponding time series data for Trait Y and selected the model that best explains the data according to a given model selection criterion (e.g. AIC or BIC). More details of this temporal-genetic causality test (TGCT) are provided in Methods.”

I encourage the authors to include some of the mathematical framework in the main text, but at the very least please provide (a) a clear definition of the specific models being evaluated;

The models being evaluated are specified in the “Temporal-genetic causality test (TGCT)” section as model M1-M5. We defined the likelihood function and BIC score for evaluating models and added the following in the Methods section (Lines 782-800):

“We used different autoregression parameters for different genotypes to account for the genetic effect and added the lagged value of one time series to represent the causal effect of one time series on the other. The parameters in each model were estimated using an ordinary linear regression. The log likelihood of X and Y are calculated as

$$\ln(\hat{L}(x)) = -\frac{N}{2} \ln(2\pi) - \frac{N}{2} \ln\left(\sum_{i,t>1} (X_{i,t} - \hat{X}_{i,t})^2\right) + \frac{N}{2} \ln(N) - \frac{N}{2}$$

and

$$\ln(\hat{L}(y)) = -\frac{N}{2} \ln(2\pi) - \frac{N}{2} \ln\left(\sum_{i,t>1} (Y_{i,t} - \hat{Y}_{i,t})^2\right) + \frac{N}{2} \ln(N) - \frac{N}{2}$$

When assessing the causal (M1: $X \rightarrow Y$) and reactive (M2: $Y \rightarrow X$) models, we calculated log joint likelihood $\ln(\hat{L}(x, y)) = \ln(\hat{L}(x)) + \ln(\hat{L}(y))$ under these two models. As the total numbers of parameters in M1 and M2 are the same, comparing $\ln(\hat{L}(x, y))$ under these two models and comparing BICs are equivalent.

One of our major goals of the temporal-genetic causality test is to identify the cis-regulators of teQTL hotspots. If we assume Trait X with a cis-eQTL linked to a teQTL hot spot, then we can restrict the model selection among causal (M1: $X \rightarrow Y$), independent (M3: $X \perp Y$), and partial causal (M4) models without considering the reactive (M2: $Y \rightarrow X$) and partial reactive (M5) models. In such cases, the three models share the same regression model for Trait X. Thus, we perform model selection based only on the regression on Trait Y. The corresponding log-likelihood was estimated as follows:

$$\ln(\hat{L}) = -\frac{N}{2}\ln(2\pi) - \frac{N}{2}\ln\left(\sum_{i,t>1} (Y_{i,t} - \hat{Y}_{i,t})^2\right) + \frac{N}{2}\ln(N) - \frac{N}{2}$$

And Bayesian Information Criterion (BIC) is defined as $BIC = \ln(N)k - 2\ln(\hat{L})$, where k is the number of parameters estimated in the corresponding model. BIC penalizes complex models. The model with the smallest BIC was identified as the model best supported by the data.”

(b) an explanation of how the underlying parameters are estimated;

The parameters were estimated by linear regression. We added the following in the Methods section (Lines 782-785):

“We used different autoregression parameters for different genotypes to account for the genetic effect and added the lagged value of one time series to represent the causal effect of one time series on the other. The parameters in each model were estimated using an ordinary linear regression.”

(c) an explanation of how the tests are applied to hotspots (is every pair of genes in the hotspot evaluated for causality?), how multiple-testing is accounted for, and how the top causal regulator is selected.

For each teQTL hot spot, we applied TGCT only to cis-trans pairs: genes with cis-eQTL linked to the hotspot and all genes with trans-teQTLs linked to the hotspot. We updated the description in the Methods Section (Lines 793-797):

“One of our major goals of the temporal-genetic causality test is to identify the cis-regulators of teQTL hotspots. If we assume Trait X with a cis-eQTL linked to a teQTL hot spot, then we can restrict the model selection among causal (M1: $X \rightarrow Y$), independent (M3: $X \perp Y$), and partial causal (M4) models without considering the reactive (M2: $Y \rightarrow X$) and partial reactive (M5) models.”

As the TGCT is based on model selection not on p-values, multiple testing correction was not applied for the test. The putative causal regulator of a teQTL hot spot is defined as the following (Lines 801-805) :

“For each teQTL hot spot, we first identified genes with cis-teQTLs linked to the hot spot as candidate causal genes, then pair these cis-eQTL genes with all genes with trans-eQTLs linked to the hot spot for the causality test. The cis-eQTL genes with the number of causal relations significantly more than expected by chance (the cutoff value for defining a teQTL hot spot) were selected as the putative key regulators of the eQTL hot spot.”

In the main text, we added the following sentence (Lines 289-290):

“For each teQTL hot spot, candidate causal genes are defined as genes with cis-teQTLs linked to the teQTL hot spot.”

And we also added the following sentence in the main text (Lines 294-296):

“The top causal regulators for each teQTL hot spot were ranked based on the number of causal relationships that the regulator had.”

2. I do not see any simulation results presented for the TGCT, nor any clear intuition on how the test is expected to perform relative to previous methods or when the underlying models are

misspecified. Additional simulations are needed to make these comparisons and show that the method actually has power at the current sample size and data complexity.

We thank the reviewer for the suggestion. We performed multiple simulation studies across a wide range of parameters. Our results suggests that the TGCT is powered to distinguish causal (M1) and reactive (M2) models with the sample size of 100 (Supplementary Figures 6&7). When leveraging genetic information and focusing only on cis-trans gene pairs, we showed that the TGCT was robust in selecting the correct models among possible models (Supplementary Figures 8-11).

We added the following section to the Results section (Lines 204-232):

“Simulation studies for evaluating the temporal-genetic causality test (TGCT)

To evaluate the performance of the temporal-genetic causality test, we simulated pairs of time series data according to causal, independent, or partial causal models specified in the Method section (detailed in Methods). We applied TGCT only to the pairs in which both time series Traits X and Y had temporal-genetic associations (MPTGA $p < 10^{-6}$) to the tested locus. First, we simulated pairs of traits according to the causal model (M1), then assessed whether the causal (M1: $X \rightarrow Y$) or reactive (M2: $Y \rightarrow X$) model fit the data better. Supplementary Figure 6 shows that TGCT assigned the correct model in most cases with accuracy of 99.54%, 99.82%, 99.95%, and 99.97% for the sample size of 20, 50, 100, 150, respectively. We also calculated the log likelihood ratio of the causal model vs. the reactive model (Supplementary Figure 7). The result suggests that our model can distinguish causal and reactive models in general. When genetic information is known, we can focus on relationships between cis-regulated genes and trans-regulated genes instead of testing all possible pairs. In the rest of tests, we assumed Trait X had a cis-teQTL so that we can simplify our tests without considering models M2 and M4. When comparing models M1, M3, and M4, we needed to model only Trait Y without explicitly modeling Trait X. For pairs simulated under the causal model (M1), TGCT identified the causal model as the best model in all cases across a wide range of strength of autoregressive and causal effects (Supplementary Figure S8), and the BIC differences between the causal model and other models are shown in Supplementary Figure S9. For pairs simulated under the independent model (M3), TGCT identified the independent model as the best model in most cases with accuracy of 95.8%, 97.7%, 97.7%, and 98.9% for the sample size of 20, 50, 100, and 150, respectively, across a wide range of parameters (Supplementary Figure S10). Simulations under the partial causal model (M4) were complicated as there were three parameters to vary to represent strength of genetic and causal effects (Supplementary Figure S11). TGCT identified the partial causal model as the best model in all cases except when genetic or causal effect was close to zero. For example, when β_2 is close to zero, the partial model (M4) is converted to the independent model (M3). In such cases, TGCT identified the independent model as the best model. When both β_{10} and β_{11} are close to zero, the partial model (M4) is converted to causal model (M1) and TGCT identified the causal model as the best model in such cases.”

The TGCT is similar to the cross-lagged mediation model (Selig JP and Preacher KJ, 2009). The main difference is that TGCT explicitly models genetic regulation in causality tests. In the cross-lagged mediation model, the causal regulator is modeled by an autoregressive model.

When we limited our causality tests to only cis-trans pairs, there was no need to explicitly model the causal regulators by an autoregressive model so that it can take a more flexible form. The cross-lagged mediation test can't be applied to genetic causality test as genotype is static and does not change over time. Instead of comparing the TGCT with the cross-lagged mediation test, we compared an autoregressive model and our MPTGA in detecting genetic associations and showed that the MPTGA was more powerful in identifying teQTLs (Supplementary Figures 4). We added the following in the Results section (Lines 193-197):

“The AR method was not included in Figure 2 as it performed worse than other methods across all conditions (Supplementary Figures 4). We also compared the power of these methods with different sample sizes, the pattern was similar as shown in Figure 2 with MPTGA performing the best when auto-correlation was high (Supplementary Figure 4).”

As the authors point out, previous work in causal inference has often overstated the statistical power to make robust statements about causality so this is a special concern.

The accuracy of the previous causality test based on static data is limited by the number of F2 strains available so that the power is limited to distinguish which gene is a true causal regulator among correlated genes co-localizing at a locus. The MPTGA has narrower confidence interval than the static method (Table 1) so that the number of candidate regulators with cis-teQTL at a locus is potentially smaller. In the time series data, there was time dimension to break correlation relationships among co-localized genes so that the TGCT has more power to distinguish which gene is true causal regulator among correlated genes co-localizing at a locus. We updated the following to the Discussion section (Lines 387-393):

“the MPTGA method was shown to detect more biologically relevant teQTL hot spots, along with tighter eQTL confidence intervals compared to the static method (Table 1) which may lead to fewer candidate causal regulators to consider in inferring key causal regulators underlying each eQTL hot spot. We also developed the causal inference test, TGCT, that simultaneously considers time series data and genetic data to systematically infer causal relationships among traits. Temporal-genetic data together has more power to distinguish which gene is true causal regulator among correlated genes co-localizing at a locus than the static method.”

Some discussion of how TGCT is expected to perform when the underlying causal model is more complex than those considered would also improve the interpretation.

There are two levels of data and model complexity. First, there are missing data points in data sets. We compared different temporal-genetic association methods when there were missing data, and showed that the MPTGA method was more robust than other tested methods (Supplementary Figure 5). We added the following to the Result section (Lines 197-202):

“To assess a model's robustness when there are missing data (detailed in Methods), we randomly dropped data points in simulated time series at various rates and applied the above methods to the data sets with missing data. The performances of the MPTGA, Regression, and AR methods

were not sensitive to missing data while the performances of the Union, Fisher, and MANOVA methods decreased as the missing data rate increased (Supplementary Figure 5).”

Second, a causal relationship may be in more complex form, such as a higher-order (time lagged) linear regressive model, a higher-order (time lagged) polynomial regressive model, or a model with multiple regulators. In the current TGCT method, we based on first-order autoregressive model. When longer time series data are available, instead of using higher-order autoregressive models, we can use multivariate polynomial functions to model causality tests. We added the following in the Discussion section (Lines 517-539):

“A number of limitations need to be addressed to further enhance the dynamic eQTL analysis. For example, one potential drawback in the TGCT test is that we explicitly modeled the causal variable in an autoregressive form, which performed less effectively in identifying genetic effects (Supplementary Figure 4). This also leads to the results that the proposed temporal-genetic association test (MPTGA) and the causality test (TGCT) are based on two different forms of models instead of a unified function. If long time series data are available, a more flexible model can be used to unify the approaches used in temporal-genetic association and causality tests. Specifically, the models can be specified as follows: given Trait X with cis-eQTL, $X_{i,t} = \delta_{i0}f_{x0}^{t-1}(t) + \delta_{i1}f_{x1}^{t-1}(t)$, and Trait Y with trans-eQTL, then the three possible models of the causal relationships between them can be rewritten as the following

$$\text{M1 (causal model): } Y_{i,t} = f^{t-1}(t) + \beta_2 X_{i,t-1} + \varepsilon_t$$

$$\text{M3 (independent model): } Y_{i,t} = \delta_{i0}f_0^{t-1}(t) + \delta_{i1}f_1^{t-1}(t) + \varepsilon_t$$

$$\text{M4 (partial causal model): } Y_{i,t} = \delta_{i0}f_0^{t-1}(t) + \delta_{i1}f_1^{t-1}(t) + \beta_2 X_{i,t-1} + \varepsilon_t,$$

Where $f_0^{t-1}(t)$ and $f_1^{t-1}(t)$ correspond to polynomial fitting functions using previous time points of each genotype, respectively; $f^{t-1}(t)$ corresponds to a single polynomial fitting function using previous time points of both genotypes. Then, we can test for temporal-genetic causality using the same model selection approach as described in the TGCT method. We can set the degrees of polynomial functions $f^{t-1}(t)$, $f_0^{t-1}(t)$, and $f_1^{t-1}(t)$ as the same as the polynomial function $f(t)$ used in MPTGA. In this study, predictions based on previous time points through the curve fitting are not possible due to limited time points available. If the number of time points is not large enough for the unified model described above, but larger than the size in our current study, we can make TGCT more flexible by using higher order autoregressive models instead of first-order autoregressive models.”

3. The applicability of this work to other studies could be significantly increased by demonstrating some general trends in statistical power. In particular, how well do the proposed methods perform if some time points are randomly dropped from the analysis? How many of the te-QTLs are still identified? And the same for dropping individuals from the analysis.

We performed simulation studies with different number of samples, and showed that the relative performance among different methods was similar across different sample sizes (Supplementary Figure 5).

We also performed simulation studies to evaluate method performances when time points were randomly dropped from the analysis. We randomly dropped time points with missing rate varying from 0.02 to 0.1. For the Union, Fisher, and MANOVA methods, missing data were excluded in calculation. For MPTGA and Regression methods, we first fit cubic curves for each genotype without considering samples with missing data, then imputed the missing time points using the estimated mean or the fit cubic curve for each genotype. We applied association tests on the imputed data. The results indicate that the MPTGA, Regression, and AR methods were not sensitive to missing data while the accuracies of the Union, Fisher, and MANOVA methods slightly decreased (Supplementary Figure 5). We added the result in the Results section mentioned above and added the following to the Methods section (Lines 827-836):

“Robustness of temporal-genetic association methods

We assessed the robustness of temporal-genetic association methods by randomly dropping data points in the simulated time series with data missing rate varying from 0.02 to 0.1. For methods that involve fitting a curve to the data within each genotype, i.e. MPTGA, regression and AR, the samples with missing time points were masked first, then each method was applied on the remaining samples (corresponding forms of curves fitted to the remaining data), then the missing time points were imputed based on the fitted curves and the temporal-genetic association methods were applied to the imputed data. For the other methods that do not fit curves to the data, i.e. union, Fisher and MANOVA, the samples with missing data were masked first and each method was applied to the remaining data.”

The discussion claims that 8 time points and 200 individuals are needed for human data; these numbers are very important but it's not at all clear how they have been derived.

We expanded the rationales for the suggested numbers and added the following in the Discussion section (Lines 559-573):

“Comparing with the yeast experiments where all experimental conditions are carefully controlled to be similar, there are multiple confounding factors that may contribute to variations to a human time course study, such as age, amount of sleep or physical exercise, food or drug taken, and diseases. Also genetic architecture of human is more complex than that of yeast. Blood is the most accessible human tissue for temporal-genetic studies. We previously studied genetic regulation of human blood transcriptome at static state [11] and in time series [24, 46]. A large amount of trans-eSNPs for human blood transcriptome were identified using 1,002 subjects [11]. Only cis-eSNPs but no trans-eSNP was identified with 40 subjects[46], suggesting it was under powered for detecting trans-eSNPs. An eQTL study in Japanese population[47] indicates that trans-eSNPs can be detected with 76 subjects. To identify more trans-eSNPs, more subjects are needed in genetic studies. We previously showed that it is possible to infer temporal causal relationships in transcription regulation of human blood transcriptome using 7 time points[24]. To effectively apply the temporal-genetic association and causality tests to a human study, we estimate that at least 8 time points and 200 individuals are needed.”

Reviewer #2 (Remarks to the Author):

The paper is focused on the analysis of genetic associations and genetic causality analysis using

temporal gene expression data. Specifically, it suggests a methodology for temporal genetic association called MPTGA and a methodology for temporal genetic causality called TGCT. The methods are demonstrated using synthetic data and data of rapamycin gene expression response across a genotyped yeast F2 population.

Major comments

1. Temporal association analysis: novelty and comparisons. The proposed association method is closely related to Ma et al. In fact the paper use the same likelihood ration test and the only difference is in the way the MLE for the parameters is calculated. In the introduction the authors claim to develop a novel methodology for temporal association, but in light of the close relation to Ma et al, the novelty of the method should be clearer stated (also in introduction) and a direct comparison with Ma et al using synthetic data is required to demonstrate superiority.

The Ma et al. paper [9] assumed the phenotype across different time points follow a logistic curve, and the MLE was based on a logistic-mixture model. In our paper, the gene expression traits followed very diverse shapes across time points (Supplementary Figure 2) which could be modeled well with a logistic curve. We model the dynamic process (the mean vectors in the multivariate normal distribution) using a polynomial form instead of the logistic curve used in Ma et al.[9]. Therefore, the log-likelihood in our paper and in Ma et al[9] take on different forms, and we used different methods to obtain the MLE of the parameters.

The synthetic data generated in the simulation studies were based on representative gene expression patterns extracted from the yeast time series dataset. The gene expression patterns (as shown in Figure S2) clearly deviate from a logistic growth curve used in Ma et al. [9]. It is unfair to show superiority of our MPTGA method over the Ma et al. method using these simulated data. Our MPTGA and the Ma et al method are similar, but not directly comparable. A polynomial form is more appropriate in fitting complex patterns. However, as more parameters need to estimate, more time points/larger sample size are needed. We added the novelty of the method and elaborate on the difference with Ma et al. [9] both in the introduction and in the main text (Line 93-97, 133-136).

“Previous method [25] proposed to model growth-related temporal traits using a multivariate normal distribution and assumed that the mean vectors followed a logistic growth curve. In the context of temporal gene expression traits, the trajectories are usually much more complex and thus require more flexible fitting options.”

and

“Instead of assuming the mean vectors of the multivariate normal distribution follow a logistic growth curve as in Ma *et al* [25], we model the mean vectors of the expression trajectories using a polynomial cubic function which is able to capture diverse types of temporal responses.”

2. Temporal Causality analysis: novelty and comparisons. The temporal causality model appears to be the same as the approach of several publications about mediation analysis (e.g., [2,3,4,5]). It seems that the only novelty is the practical translation to genetics. If there are no substantial differences, this should be emphasized. If there is a substantial difference, a comparison to the existing methods in Refs [2-5] should be included. Differences from this literature should be also discussed in the introduction section.

The major difference between the temporal genetic causality test and the cross-lagged mediation methods is the way how the genetic effect is incorporated into the model. In the cross-lagged mediation models, each variable/mediator is modeled by an autoregressive model. Genotype information is static, and does not change across time so that genotype information can't be modeled in the cross-lagged mediation model. We add the following in the Introduction section (Lines 155-157):

“Several mediation models for longitudinal data were developed based on Granger causality [28], but no model takes genetic data into consideration.”

3. The code for MPTGA and TGCT should be released on a repository such as cRAN. All the codes will be available at our website <http://research.mssm.edu/integrative-network-biology/Software.html>.

4. Statistical validity of the association with MPTGA. It is important to demonstrate that the p-values derived from the chi-sq approximation (lines 537-543) are calibrated. QQ-plots and inflation parameter are required in both real and synthetic data. The large number of associations in the case of ribosomal proteins suggest spurious associations of responding genes and therefore it is especially important to show that the ribosomal modules are not only due to statistical inflation (see also comments 5,6,8).

We thank the reviewer for the suggestion. We compared QQ plots for simulated data, permuted data, and real data. It is true that p-values for simulated data (Supplementary Figure 18A) and permuted data (Supplementary Figure 18B) were inflated. The results emphasize the importance of using results of permuted data to adjust significance of results (Supplementary Figure 18D) instead of using the p-values of the real data (Supplementary Figure 18D) directly. We added the following to the Discussion section (Lines 466-478)

“We also checked statistical validity and potential inflation of p-values of MPTGA (detailed in Methods). First, we simulated a set of gene expression traits and genotypes. As they were simulated independently, no gene association was expected. The QQ plot for the simulated data (Supplementary Figure 18A) suggests that p-values of MPTGA are slightly inflated. We then generated permuted data from the yeast F2 data set by permuting strain labels so that the genetic structure is intact. As there are LD structures in the genetic structure, the QQ plot for the permuted data (Supplementary Figure 18B) is slightly different from the QQ plot for the simulated data. The QQ plot for the yeast F2 data (Supplementary Figure 18C) is significantly different from the plots for simulated and permuted data. The QQ plot comparing results of the

real data and permuted data (shown in Supplementary Figure 18D) indicates that the result of the real data was significantly different from the result of the permuted data. These results together suggest that p-value itself is not accurate and it is better to use FDR values to control errors.”

Figure S18 QQ plots of the p-values of the MPTGA test. **A)** Simulated data; **B)** permuted data; **C)** real data; **D)** p-values for real data v.s. permuted data.

5. Statistical validity of the causality analysis with TGCT. The different models are compared using BIC criteria and the best model according to the BIC score is preferred. Several recent papers show that it is preferable to perform genetic causality using more advanced tests (rather than the BIC or AIC criteria), such as Ref [7] (Neto et al). The distribution of likelihood ratios between M1 versus the max of the alternative models (as in [7]) should be also evaluated empirically using reshuffling of gene expression, especially for DDR1 and for the ribosomal genes, since the theoretical assumptions do not necessarily hold in the real scenario. Note that Neto et al. is also our work, which was done when Dr. Neto was a postdoc in Zhu lab. As Neto et al stated, the power of Vuong’s test based causality test is low and the test is conservative. The Vuong’s test based method is suitable for testing trans-trans pairs whose number is huge, but not suitable for testing cis-trans pairs.

We agreed with the reviewer that not only the best fit model is important, but also the posterior probability of the best model is important. In our simulation tests, we reported accuracy of model selection as well likelihood ratios of M1 vs M2 (Supplementary Figure 7) and BIC difference between M1 and other models (Supplementary Figure 9). The distribution for the sample size of 100 (Supplementary Figure 9C) is shown below.

Figure S9: The distribution of the BIC difference between the causal model and other models based on simulated data used in Figure S8 for different sample sizes $N=20, 50, 100,$ and 150 (A-D, respectively).

In our TGCT, we require both Traits X and Y with teQTLs linked to the same locus. Thus, we can't assess likelihood ratio distribution based on permuted data as the permutation destroys the genetic association required for the TGCT. For the inferred causal pairs, we reported the BIC difference of M1 vs. the second best model in Supplementary Figure 12. For causal pairs with RRD1 as a causal regulator, the distribution of BIC difference is shown in Supplementary Figure 13, strongly supporting causal model M1 over other models.

Figure S12 The distribution of the BIC difference between the causal model and other models based on yeast F2 data for all causal pairs in Supplementary Table 11.

Figure S13 The distribution of the BIC difference between the causal model and other models based on yeast F2 data for all causal pairs with *RRD1* as a causal regulator.

6. Prior work (see for example Ref [8]) indicate that association hotspots are mainly due to spurious associations. This may be the case here in light of the large number of associations of ribosomal genes. eQTL maps should be demonstrated (as in [8]) and control for spurious associations should be applied if necessary.

The NICE method aims to identify confounding factors that potentially correlate with genotypes. Our time series data set is more complex than standard data sets used in genetic gene expression studies as there is a time component. Thus, the NICE method can't be applied to our data set directly. Similar to what Joo et al (Ref 8) did, we compared the QTL hot spots at T0 with previous results and all large QTL hot spots overlap with ones based on independent data for the same yeast F2 cross (Table 1), suggesting that there was no systematic confounding factor in the data set. Ribosomal genes were enriched in QTL hot spots at Static state (Yvert et al, 2003; Smith and Kruglyak, 2008; Zhu et al, 2008). As rapamycin targets ribosome biosynthesis, it is expected that a large number of ribosomal genes associate with teQTLs.

7. The relevance to human and mouse data is not clear. Both methods rely on binary SNP of 0/1 values. Heterozygosity is not handled, and extension to inbred mice populations (such as CC and DO) that rely on multiple parental strains is not clear. Relevance and possible extensions should be discussed.

When applying the MPTGA and TGCT to diploid systems in which there are three possible genotypes, 00/01/11 (or 0/1/2), at each SNPs, we can apply these methods directly to detect dominant/recessive effects. To detect full genetic effects, the genetic association test can be

expressed as $y_i(t) = \delta_{i0} \sum_{k=0}^K \beta_{k0} t^k + \delta_{i1} \sum_{k=0}^K \beta_{k1} t^k + \delta_{i2} \sum_{k=0}^K \beta_{k2} t^k \varepsilon_i$ for a given trait Y, then the reduced model H_0 (single gene expression trait curve)

$$H_0 : \beta_{k0} = \beta_{k1} = \beta_{k2} \text{ for all } k$$

can be compared against the full model H_1 (different gene expression trait curve for different genotypes):

H_1 : at least one of the equalities does not hold

to test the hypothesis of the existence of eQTL at a locus by estimating these parameters with an MLE procedure and performing likelihood ratio test as we described in the Method section. Similar generalization can be applied to the TGCT. We added the above to the Discussion section (Lines 504-516).

8. Recent prior work [1] (Brodt et al.) has already applied temporal eQTL analysis on the same yeast dataset. It seems that most of the identified eQTLs here have been already published in the previous work. A clear comparison should be included (which modules are the same? which ones are novel and which ones are missing?). From a brief comparison, I noticed that the main contribution here is the identification of associations in ribosomal genes. However, this result may be an artifact, as none of the ribosomal genes were previously identified; this may occur due to inflation of scores in the highly responding ribosomal genes.

We respectively disagree with the reviewer that the effect on ribosomal genes is likely an artifact. Biologically, rapamycin regulates cell growth and ribosome biogenesis. It would be surprised if there is no temporal-genetic effect on ribosome biogenesis. Our results suggest that there are many ways to regulate ribosome biogenesis and in turn affect cell growth.

In term of novelty, our method takes all time points into consideration and fits a curve first, then compares whether curves for different genotype are different. The DyVER (Brodt et al. 2014) is an advanced meta-analysis method, which analyzes genetic effect at each individual time point first, then discretize genetic effects at different time points into two states. Our simulation results indicate that the MPTGA method is more sensitive than methods considering individual time points separately when variances were not independent (Figure 2). More importantly, when there are missing data, there is no performance degradation for methods considering all time points together, but there is clear performance degradation for methods considering individual time points separately (Supplementary Figure 5). Comparing results of Brodt et al and our results, all modules identified in Brodt et al.'s Fig5A were also identified by all methods (Table 2) except the ChrIII MATalpha hot spot. Both the T0, and Fisher methods identified the ChrIII MATalpha hot spot, but the number of traits linked to the locus was less than the cutoff for the MPTGA (Figure 3 shows that there is a small peak at the locus). The Brodt et al.'s Fig. 5A indicates that the genetic effect at the locus was not dynamic. We added the above to the Discussion section (Lines 484-495).

The validity of ribosomal genes result needs to be addressed via (i) clear visualization of the different dynamics in ribosomal genes with different genetic backgrounds (e.g., as in Figure 4B in Ref [1]),

Figure 4A to include means of RPPA2 expression levels of each genotype. The time series plot for individual stain's RPPA2 expression level is shown in Supplementary Figure 14.

Figure S14: Individual expression pattern of RPP2A which was linked to the teQTL hot spot chrIX:70,000.

(ii) showing that it is not due to inflation or confounding effects (as mentioned in Q. 4 above)

We included differences of BICs for pairs regulated by RDD1 as Supplementary Figure 13 (showed above).

(iii) discuss why the ribosomal associations were not previously identified. As it stands, the interpretation of the ribosomal genes associations—a major part of the actual novelty—is a major concern.

From Fig4A, it is clear that the largest difference was at the last time point, but the trend across time point was consistent. Considering the last time point alone, the Wilcoxon rank-sum test p-value was <0.01 (Figure 4D), but not significantly at genome level. The DyVER method (as well as the method proposed by Francesconi and Lehner[47]) which aims to group genetic effects at each individual time point into two discrete states is unlikely to work well in the cases with moderate genetic effect changes over time. This also highlights the advantage of the MPTGA method that simultaneously takes all time points into consideration. The above is added the Discussion section (Lines 495-503).

Minor comments:

 9. Two recently published temporal association methods are Refs [1] and [6]. It is advantage to compare to these methods.

Brodt et al and Francesconi&Lehner applied a similar approach, first to identify genetic effects at each individual time points, and then to compare whether the genetic effects are similar or not.

Their approaches are fundamentally different from the MPTGA method which takes all time points into consideration simultaneously. We add the two references in the Discussion section (showed above).

10. How the methodology handles intraindividual change over time. Is it possible to discriminate between time-specific causal effects from sustained causal effects?

For temporal-genetic association, we modeled time series data in an autoregressive (AR) model to capture time-specific causal effect (detailed in Methods; Supplementary Figure 4). The TGCT incorporate both genetic and temporal causal effect.

Figure S4: Performance comparison among different temporal genetic association methods under different strength of inter time point correlation (auto-correlation) based simulated data. Area under the curve (AUC) was used to assess model accuracy. In each simulated dataset, $\rho_i \sim N(\rho, 0.02)$, $i = 1, 2, \dots, 10000$. The comparison is shown for different sample sizes N=20, 50, 80, and 100 to demonstrate the general trend in statistical power of each method.

11. The model assumes a 2-time points lag. In the literature a longer lag is also considered. We only used first-order autoregressive model in the TGCT due to only short time series data available. If long time series data is available, more flexible models can be applied to detect genetic causal effects. We added the following to the Discussion section (Lines 517-539).

“A number of limitations need to be addressed to further enhance the dynamic eQTL analysis. For example, one potential drawback in the TGCT test is that we explicitly modeled the causal variable in an autoregressive form, which performed less effectively in identifying genetic effects (Supplementary Figure 4). This also leads to the results that the proposed temporal-genetic association test (MPTGA) and the causality test (TGCT) are based on two different forms of models instead of a unified function. If long time series data are available, a more flexible model can be used to unify the approaches used in temporal-genetic association and causality tests. Specifically, the models can be specified as follows: given Trait X with cis-eQTL, $X_{i,t} = \delta_{i0}f_{x0}^{t-1}(t) + \delta_{i1}f_{x1}^{t-1}(t)$, and Trait Y with trans-eQTL, then the three possible models of the causal relationships between them can be rewritten as the following

$$\text{M1 (causal model): } Y_{i,t} = f^{t-1}(t) + \beta_2 X_{i,t-1} + \varepsilon_t$$

$$\text{M3 (independent model): } Y_{i,t} = \delta_{i0}f_0^{t-1}(t) + \delta_{i1}f_1^{t-1}(t) + \varepsilon_t$$

$$\text{M4 (partial causal model): } Y_{i,t} = \delta_{i0}f_0^{t-1}(t) + \delta_{i1}f_1^{t-1}(t) + \beta_2 X_{i,t-1} + \varepsilon_t,$$

Where $f_0^{t-1}(t)$ and $f_1^{t-1}(t)$ correspond to polynomial fitting functions using previous time points of each genotype, respectively; $f^{t-1}(t)$ corresponds to a single polynomial fitting function using previous time points of both genotypes. Then, we can test for temporal-genetic causality using the same model selection approach as described in the TGCT method. We can set the degrees of polynomial functions $f^{t-1}(t)$, $f_0^{t-1}(t)$, and $f_1^{t-1}(t)$ as the same as the polynomial function $f(t)$ used in MPTGA. In this study, predictions based on previous time points through the curve fitting are not possible due to limited time points available. If the number of time points is not large enough for the unified model described above, but larger than the size in our current study, we can make TGCT more flexible by using higher order autoregressive models instead of first-order autoregressive models.”

References

- [1] Dissecting Dynamic Genetic Variation That Controls Temporal Gene Response in Yeast. A Brodt, M Botzman, E David, I Gat-Viks. PLOS Comp Biol 2014.
- [2] Scott E. Maxwell, David A. Cole & Melissa A. Mitchell (2011) Bias in Cross-Sectional Analyses of Longitudinal Mediation: Partial and Complete Mediation Under an Autoregressive Model, Multivariate Behavioral Research, 46:5, 816-841, DOI:10.1080/00273171.2011.606716

- [3] James P. Selig and Kristopher J. Preacher (2009). Mediation Models for Longitudinal Data in Developmental Research. *Research in Human Development*, 6(2–3), 144–164, 2009
- [4] Magnusson Hanson LL, Peristera P, Chungkham HS, Westerlund H (2016) Longitudinal Mediation Modeling of Unhealthy Behaviors as Mediators between Workplace Demands/Support and Depressive Symptoms. *PLoS ONE* 11(12): e0169276. doi:10.1371/journal.pone.0169276
- [5] AN EXAMINATION OF STATISTICAL METHODS FOR LONGITUDINAL MEDIATION MODELING, dissertation by BIRMINGHAM, ALABAMA, 2011.
- [6] Francesconi M, Lehner B (2014) The effects of genetic variation on gene expression dynamics during development. *Nature* 505: 208–211.
- [7] Modeling causality for pairs of phenotypes in system genetics EC Neto, AT Broman, MP Keller, AD Attie, B Zhang, J Zhu, BS Yandell *Genetics*, 193: 1003-1013, 2013. DOI: <http://dx.doi.org/10.1534/genetics.112.147124>
- [8] Effectively identifying regulatory hotspots while capturing expression heterogeneity in gene expression studies. Jong Wha J Joo, Jae Hoon Sul, Buhm Han, Chun Ye and Eleazar Eskin *Genome Biology* 2014. DOI: 10.1186/gb-2014-15-4-r61

Reviewer #3 (Remarks to the Author):

The manuscript by Zhu and colleagues studies the issue of mapping eQTL in time-series expression data (temporal eQTL or teQTL), and proposes to combine the benefits of genetic perturbations and time series information to reconstruct causal relationship between genes. Most eQTL studies were performed on single conditions, but as the authors argued, there is clearly a need of exploring dynamic transcriptome response. Thus, the teQTL methodology described in this work is very interesting, and the results seem to demonstrate the advantage of the method they developed (MPTGA). The authors also studied the problem of identifying causal regulators of gene expression traits, using the idea of Granger causality. However, I have some serious doubt of this part of their work.

Major comments

The authors evaluated their teQTL method, MPTGA, against simpler alternative such as union method and Fisher's meta-analysis. But mapping teQTL can be viewed as a problem of testing association of a SNP with multiple phenotypes. One can simply view expression level at each time point as a trait, and test all traits (i.e. time points) jointly. This problem of cross-phenotype association has been extensively studied before, e.g. see PMID:23752797 for a review. Thus it would be appropriate to evaluate the performance of these related methods in the teQTL setting. In particular, a common multivariate association method is based on MANOVA (or its variants). Basically, suppose there are K traits, one has $Y_k = \beta_k G + \epsilon$, where G is genotype of the test SNP, and Y_k the k -th trait. The problem is to test $H_0: \beta_1 = \dots = \beta_K = 0$.

We added the comparison with MANOVA in our comparison (Figure 2, Supplementary Figures 4) and updated results accordingly. MPTGA performed better than MANOVA across all

conditions. And importantly, MPTGA is robust when there are missing data while MANOVA's performance decreases when the missing data rate increases (Supplementary Figure 5).

Figure S4: Performance comparison among different temporal genetic association methods under different strength of inter time point correlation (auto-correlation) based simulated data. Area under the curve (AUC) was used to assess model accuracy. In each simulated dataset, $\rho_i \sim N(\rho, 0.02)$, $i = 1, 2, \dots, 10000$. The comparison is shown for different sample sizes $N=20, 50, 80,$ and 100 to demonstrate the general trend in statistical power of each method.

Figure S5: Performance comparison of MPTGA and other methods when there are time points randomly dropped from the time series data with missing rate varying from 0.02 to 0.1. For methods that involve fitting a curve to the data within each genotype, i.e. MPTGA, regression and AR, the samples with missing time points were masked first, then each method was performed on the remaining samples (corresponding forms of curves fitted to the remaining data). Next, the missing time points were imputed using the fitted curves. Finally, each method was applied on the imputed data. For the other methods that do not fit curves to the data, i.e. union, Fisher and MANOVA, the samples with missing data were masked first and each method was applied on the remaining data.

The authors compared MPTGA with three other methods in yeast data and showed a higher performance of MPTGA (Table 1). One question is that what is the extent of the overlap among these methods. It is possible that MPTGA finds most of the teQTL identified by other methods, but gain some extra ones. But it is also possible that MPTGA captures somewhat different aspects of data than other method, and thus the overlap of teQTL from different studies would be relatively small. My guess is that MPTGA is good at capturing the overall "shape" of the time series data, but others may be more sensitive to change of expression at only one time point. Among 11 teQTL hot spots defined by MPTGA, 8 were identified by the Union and Fisher methods, too (Table 2). And the numbers eQTLs at these hot spots were similar as well. These together suggest that all methods can detect teQTLs if genetic associations were strong at least at one time point. As an example showed in Figure 4A, the teQTLs identified by MPTGA but not the Union nor Fisher methods were likely to reflect shape of time series data rather than the absolute differences between genotypes at each time point. We add the following to the Discussion section (Lines 496-499):

“For example, the ChrIX70,000 hot spot includes gene expression with genetic effects gradually changing over time (Figure 4A). Considering the last time point alone, the Wilcoxon rank-sum test p-value was <0.01 (Figure 4D), but not significantly at genome level.”

The TGCT model (equations from line 581 to 589) looks somewhat strange to me. The authors tries to expand the Granger causality test to include genetic effects. But the way they model genetic effects is completely different from what they have done in the teQTL part. In the TGCT model, a SNP could change the coefficients in the Granger causality model, e.g. alpha_00 vs. alpha_01 and alpha_10 vs. alpha_11 in M1. But these coefficients represent the extent of auto-correlation between adjacent time points of an expression trait, while eQTL (or teQTL) measures the genetic effect on actual expression level. For example, the alpha parameters here (say M1) do not change with time, while the eQTL effects are generally time-dependent (say a SNP may affect expression only in one time point). It is difficult to reconcile these very different views of eQTL, and I do not see any evidence at all in this paper, that eQTL would behave in this way (i.e. changing auto-correlation coefficients). This may be related to a comment below about the Table 4 results.

We modeled temporal-genetic associations in an autoregressive (AR) model (see Methods for details). The AR method had an AUC above 0.8 when the sample size was 100 (Supplementary Figure 4), suggesting that the AR method can identify temporal-genetic associations. We agreed with the Reviewer that the AR method is not the best way to model temporal-genetic association in TGCT. We made the choice due to limited number of time points available. In our study, we focused on cis-trans causal relationships. After a cis-teQTL was identified by the MPTGA method, we tested only causal model (M1), independent model (M3), and partial causal model (M4) so that we don't need to explicitly model the cis-eQTL trait X in an AR model. If long time series data is available, the MPTGA model can be incorporated into TGCA. We added the following in the Discussion section (Lines 517-539):

“A number of limitations need to be addressed to further enhance the dynamic eQTL analysis. For example, one potential drawback in the TGCT test is that we explicitly modeled the causal variable in an autoregressive form, which performed less effectively in identifying genetic effects (Supplementary Figure 4). This also results that the proposed temporal-genetic association test (MPTGA) and the causality test (TGCT) are based on two different forms of models instead of a unified function. If long time series data are available, a more flexible model can be used to unify the approaches used in temporal-genetic association and causality tests. Specifically, the models can be specified as follows: given Trait X with cis-eQTL,

$X_{i,t} = \delta_{i0}f_{x0}^{t-1}(t) + \delta_{i1}f_{x1}^{t-1}(t)$, and Trait Y with trans-eQTL, then the three possible models between them can be rewritten as the following

$$\text{M1 (causal model): } Y_{i,t} = f^{t-1}(t) + \beta_2 X_{i,t-1} + \varepsilon_t$$

$$\text{M3 (independent model): } Y_{i,t} = \delta_{i0}f_0^{t-1}(t) + \delta_{i1}f_1^{t-1}(t) + \varepsilon_t$$

$$\text{M4 (partial causal model): } Y_{i,t} = \delta_{i0}f_0^{t-1}(t) + \delta_{i1}f_1^{t-1}(t) + \beta_2 X_{i,t-1} + \varepsilon_t ,$$

Where $f_0^{t-1}(t)$ and $f_1^{t-1}(t)$ correspond to polynomial fitting functions using previous time points of each genotype, respectively; $f^{t-1}(t)$ corresponds to a single polynomial fitting function using previous time points of both genotypes. Then, we can test for temporal-genetic causality using the same model selection approach as described in the TGCT method. We can set the degrees of polynomial functions $f^{t-1}(t)$, $f_0^{t-1}(t)$, and $f_1^{t-1}(t)$ as the same as the polynomial function $f(t)$ used in MPTGA. In this study, predictions based on previous time points through the curve fitting are not possible due to limited time points available. If the number of time points is not large enough for the unified model described above, but larger than the size in our current study, we can make TGCT more flexible by using higher order autoregressive models instead of first-order autoregressive models.”

The procedure for finding causal regulator of teQTL hotspots is not described clearly. I would suggest the authors to create a separate subsection in Methods to explain the procedure. Given a teQTL hotspot and its associated genes, one key question I have is: do they search all genes within the hotspot; or only genes that are cis-associated with the hotspot (from teQTL analysis). Another question is: they need to compare the number of causal relations between a cis-gene and the hotspot-associated genes with random expectation, how is this random expectation calculated? Because the hotspot-associated genes have correlated expression patterns, some extra care is needed to obtain the random distribution.

We clarified the confusion and stated that we tested only cis-trans pairs in each hot spot. As we can't permute cis-trans pairs (otherwise the temporal-genetic associations would lose), we ranked causal regulators by the numbers of causal relationships that each regulator has. For the cutoff number of causal relationships for defining key regulator, we used the same cutoff for define a teQTL hot spot. We update the main text (Lines 289-290 and 294-296).

“For each teQTL hot spot, candidate causal genes were defined as genes with cis-teQTLs linked to the teQTL hot spot.”

and

“The top putative causal regulators for each teQTL hot spot were ranked based on the number of causal relationships that the regulator had.”

We also added the following to the Methods section (Lines 802-806):

“For each teQTL hot spot, we first identified genes with cis-teQTLs linked to the hot spot as candidate causal genes, then pair these cis-eQTL genes with all genes with trans-eQTLs linked to the hot spot for the causality test. The cis-eQTL genes with the number of causal relations significantly more than expected by chance (the cutoff value for defining a teQTL hot spot) were selected as the putative key regulators of the eQTL hot spot.”

In Table 4, the authors reported a number of candidate causal cis-regulators of teQTL hotspots. The results look suspicious to me. The teQTL hotspots are not large, based on the authors description (perhaps 10 genes on average, as they have 600 bins). And I would think that most hotspots have 1-2 true causal regulators that mediate the effect of hotspots on the associated genes. But the authors reported much larger number of regulators. The main justification in the manuscript is some experimental and literature evidence of two genes ISC1 and RRD1. We

would need much stronger evidence that most of the predicted regulators are real. In light of possible statistical problems (two previous comments), I am concerned that many of them could be false positives.

Neighboring bins with numbers of eQTLs above a threshold for an eQTL hot spot were merged into one eQTL hot spot in the same way as Brem et al (Science 2002) described. The eQTL hot spot with a large number of eQTLs may cover multiple bins (Figure 3), such as the chrII:550,000 and chrXV:150,000. We then identified genes with cis-eQTLs at each eQTL hot spot as candidate genes for each hot spot. As some hot spots were wider, there were more genes in the hot spot regions. Second, for each teQTL hot spot, we paired all genes with cis-eQTLs at the hot spot and all genes with trans-eQTLs at the hot spot, then tested potential causal relationships. For a gene with a trans-eQTL at the hot spot, our test may report multiple candidate causal genes. On the other hand, for two genes (X1 and X2) with cis-eQTLs at the hot spot, they may be causal to an overlapped set of genes (Ys) with trans-eQTLs linked the hot spot, for example, $X1 \rightarrow Y$ and $X2 \rightarrow Y$. In these cases, our test can't distinguish which cis-eQTL gene is the true causal gene. Thus, multiple putative causal genes were reported for hot spots with a large number eQTLs. To distinguish which causal relationship, $X1 \rightarrow Y$ or $X2 \rightarrow Y$, is true, more data are needed, such as more F2 strains to breakdown LD structures or more time points to narrow eQTL confidence intervals. We added the following to the Discussion section (Lines 398-410).

Minor comments

Are the authors planning to make their software MPTGA widely available? It seems useful. My question is about generality of the software. Can it be used for data with an arbitrary number of times point? In the method presented, the authors fit a cubic polynomial, but to use it in another dataset, a higher degree may be needed. Is there a routine to automatically choose such parameters?

Yes, software implementing MPTGA and TGCT will be public available at our website <http://research.mssm.edu/integrative-network-biology/Software.html>. The methods can handle any number of time points (assuming that the number is not too low). The best degree of a polynomial function was determined by goodness of fit (Supplementary Figure 22).

RRD1 has two eQTL, one cis in chrIV:70K and another trans in chrIV:150K. There are teQTL hotspots in both loci. The authors said that both hotspots could act be changing RRD1 level, if this is true, the genes associated with both hotspots should have significant overlap. Is this true? There was a set of genes regulated by multiple genetic signals (Figure 5C). Yes, 78% (50 out of 64) of genes linked to chrIX:70K were also linked chrXV:150K hot spots. And genes in the hot spot chrXV:150K changed in response to RRD1 knockout (Figure 5A).

Line 249, 253: Should "Punitive" be "putative"?
We corrected the typo.

REVIEWERS' COMMENTS:

Reviewer #1 (Remarks to the Author):

The authors have thoroughly addressed all of my comments. I appreciate their attention to detail.

Reviewer #3 (Remarks to the Author):

The authors have done a great job addressing the concerns in my previous review. My only minor comment is that: the Discussion section is quite long. They have included detailed discussions on issues raised by reviewers, including myself. However, I think some of these can be shortened, e.g. the discussion about how to combine MPTGA and TGCT in a unified framework (the equations can probably be skipped).

REVIEWERS' COMMENTS:

Reviewer #1 (Remarks to the Author):

The authors have thoroughly addressed all of my comments. I appreciate their attention to detail.

We thank the encouraging comment.

Reviewer #3 (Remarks to the Author):

The authors have done a great job addressing the concerns in my previous review. My only minor comment is that: the Discussion section is quite long. They have included detailed discussions on issues raised by reviewers, including myself. However, I think some of these can be shortened, e.g. the discussion about how to combine MPTGA and TGCT in a unified framework (the equations can probably be skipped).

We thank the encouraging comment and suggestions. We shortened the Discussion section and moved details of how to create a generalized MPTGA and a unified framework for MPTGA and TGCT into the Methods.